# Co-contraction embodies uncertainty: An optimal feedforward strategy for robust motor control

**Bastien Berret**[1,2¤]*, **Dorian Verdel**[3], **Etienne Burdet**[3], **Frédéric Jean**[4]

**1** Université Paris-Saclay CIAMS, Orsay, France, **2** CIAMS, Université d'Orléans, Orléans, France, **3** Imperial College of Science, Technology and Medicine, London, United-Kingdom, **4** Unité de Mathématiques Appliquées, ENSTA Paris, Institut Polytechnique de Paris, Palaiseau, France

¤ Current address: Faculté des Sciences du Sport, bâtiment 335, 91405 Orsay Cedex, France
* bastien.berret@universite-paris-saclay.fr

**Data Availability Statement:** All relevant data are within the manuscript and its Supporting information files.

**Funding:** This work was supported in part by the French National Agency for Research (grant ANR-

## Abstract

Despite our environment often being uncertain, we generally manage to generate stable motor behaviors. While reactive control plays a major role in this achievement, proactive control is critical to cope with the substantial noise and delays that affect neuromusculoskeletal systems. In particular, muscle co-contraction is exploited to robustify feedforward motor commands against internal sensorimotor noise as was revealed by stochastic optimal open-loop control modeling. Here, we extend this framework to neuromusculoskeletal systems subjected to random disturbances originating from the environment. The analytical derivation and numerical simulations predict a characteristic relationship between the degree of uncertainty in the task at hand and the optimal level of anticipatory co-contraction. This prediction is confirmed through a single-joint pointing task experiment where an external torque is applied to the wrist near the end of the reaching movement with varying probabilities across blocks of trials. We conclude that uncertainty calls for impedance control via proactive muscle co-contraction to stabilize behaviors when reactive control is insufficient for task success.

## Author summary

This work presents a computational framework for predicting how humans modulate muscle co-contraction to cope with uncertainties of different origins. In our neuromusculoskeletal system, uncertainties have both internal (sensorimotor noise) and external (environmental randomness) origins. The present study focuses on the latter type of uncertainty, which had not been dealt with systematically previously despite its importance in everyday life. Therefore, we thoroughly investigated how random disturbances occurring with some probability in a motor task shape the feedforward control of mechanical impedance through muscle co-contraction. Here we provide theoretical, numerical and experimental evidence that the optimal level of co-contraction steeply increases with the uncertainty of our environment. These findings show that muscle co-

19-CE33-0009 to BB). The funders had no role in study design, data collection and analysis, decision to publish, or preparation of the manuscript.

**Competing interests:** The authors have declared that no competing interests exist.

contraction embodies uncertainty and optimally mitigates its consequences on task execution when feedback control is insufficient due to sensory noise and delays.

## Introduction

The co-contraction of muscles spanning a joint has long been studied in human motor control (see [1] for a review). Although metabolically costly, humans will rely on an anticipatory –possibly transient– muscle co-contraction to perform a motor task on various occasions. For instance, muscle co-contraction is increased when walking on uneven ground compared to flat ground [2]. For upper-limb movements, the use of muscle co-contraction has been well characterized in tasks involving adaptation to unstable dynamics typical of tool use [3–6]. Whether co-contraction serves to modulate the mechanical impedance of the system [3] or to make feedback control more efficient [7, 8] (for instance via scaling up gains [9, 10] or enhancing response times of muscles [11]), it does contribute to robustify motor behaviors by making them more stable, accurate and reproducible [12–14]. However, few models of neuromechanical control provide a principled account of muscle co-contraction. The classical stochastic optimal control theory does not predict co-contraction well as it would constitute a waste of energy compared to the efficient feedback control [15, 16]. Yet, there are significant noise and delays in the central nervous system (CNS) [17, 18] so that pure feedback control may not be a viable strategy for the task at hand [19, 20]. Other models have proposed that feedforward control can generate robust motor behaviors, especially with variable impedance systems like the human neuromusculoskeletal system [21–23]. When considering the noise and delays in feedback loops, feedforward co-contraction can even constitute a minimum-effort strategy to reliably perform the motor task [24, 25]. Interestingly, there is evidence that humans can internally represent the effects of mechanical impedance on movement, even without physical contact [26, 27]. Therefore, it is plausible that the CNS can plan mechanical impedance via feedforward muscle co-contraction depending on the task's characteristics. One of these fundamental characteristics is task uncertainty [28].

Although most computational models focus on the uncertainty arising from within the CNS, such as motor noise, feedback-driven fluctuations or command timing volatility [15, 29–31], uncertainty can also come from the environment and trigger motor adaptations (e.g., [32–36]). For instance, when exposed to a force field that randomly switches on and off between consecutive trials, humans tend to co-contract the muscles of the relevant joints in anticipation as a response [37, 38]. Incidentally, the very estimation of mechanical impedance requires the application of unpredictable disturbances to human limbs (e.g., [39]). Since humans usually adapt to such disturbances by increasing their mechanical impedance, this illustrates how uncertainty and impedance are intricately connected quantities. Therefore, the development of computational models to predict how the CNS should modulate muscle co-contraction as a function of task uncertainty will shed light on this ubiquitous motor strategy.

Here, we extend our previous framework of stochastic optimal open-loop control [22, 23, 40] to handle both internal and external types of uncertainty. Importantly, this framework can be applied to the nonlinear neuromusculoskeletal system. This partly comes from the restriction to open-loop control, which allows us to derive efficient methods for computing the optimal level of feedforward co-contraction given the task uncertainty, by leveraging the tools of deterministic optimal control. Our theoretical analysis predicts the existence of a logarithmic relationship between environmental uncertainty and muscle co-contraction, so that co-

contraction should steeply increase with the degree of task uncertainty. This is tested in an experiment with human participants, which confirms the plausibility of the theory.

## Results

### Uncertain stochastic optimal open-loop control

The proposed framework assumes a nonlinear control system subjected to uncertainties arising from the sensorimotor noise in the CNS and the randomness of the environment. These sources of uncertainty differ in their nature and are therefore modeled distinctly. More precisely, we shall consider the nonlinear stochastic differential equation in Itô's sense [41]

$$d\mathbf{x}_t = \mathbf{f}(\mathbf{x}_t, \mathbf{u}(t), t; \boldsymbol{\xi})\, dt + \mathbf{G}(\mathbf{x}_t, \mathbf{u}(t), t)\, d\mathbf{w}_t, \tag{1}$$

where $\mathbf{x}_t \in \mathbb{R}^n$ is the state vector (including position, velocity etc.), $\mathbf{u}(t) \in \mathbb{R}^m$ the control vector (e.g., muscle activations or torques), $\boldsymbol{\xi}$ a $p$-dimensional random vector modeling the environmental uncertainty (discrete or continuous with mean $\boldsymbol{\mu}$ and covariance $\Sigma$) and $\mathbf{w} = \{\mathbf{w}_t\}_{t\in\mathbb{R}^+}$ a multi-dimensional Wiener process modeling the internal uncertainty in the CNS, i.e. motor noise [16, 29, 42]. The drift $\mathbf{f}$ and diffusion $\mathbf{G}$ are smooth nonlinear functions. As a special case, the drift may exhibit an affine dependence on $\boldsymbol{\xi}$, that is $\mathbf{f}(\mathbf{x}_t, \mathbf{u}(t), t; \boldsymbol{\xi}) = \mathbf{g}(\mathbf{x}_t, \mathbf{u}(t), t) + \mathbf{H}(\mathbf{x}_t, \mathbf{u}(t), t)\boldsymbol{\xi}$. This illustrates that the random variable $\boldsymbol{\xi}$ can trigger any arbitrary force field that depends on the state, control and/or time. In this framework, each trajectory $\mathbf{x}_t$ generated by the control $\mathbf{u}(t)$ depends on the fluctuations of the random variable $\boldsymbol{\xi}$, the Wiener process $\mathbf{w}$ and the random initial state $\mathbf{x}_0$ (with mean $\mathbf{m}_0$ and covariance $\mathbf{P}_0$). Knowledge about this initial state could originate from a state estimation process –not modeled here– and, therefore, it also introduces some uncertainty due to sensorimotor noise. Throughout the paper, we will use the notation $\mathbf{x}_t$ to distinguish the stochastic process solution to a stochastic differential equation (SDE) from the random time function $\mathbf{x}(t)$ solution to an ordinary differential equation (ODE). The latter occurs when the diffusion term $\mathbf{G}$ is zero. Furthermore, since we focus on the role of feedforward motor commands, we explicitly assume that the control is open-loop throughout the paper, and write it as the deterministic function $\mathbf{u}(t)$ for $t \in [0, T]$ where $T$ is the length of the planning time horizon. Accordingly, our model focuses on the effects of the intrinsic viscoelastic properties of muscles and their pre-reflex/built-in responses, which contribute to modulating limb mechanical impedance [43, 44]. It does not take into account high-level feedback responses that involve state estimation processes, which are central to stochastic optimal feedback control [15, 45] (see the Discussion).

We thus assume that the motor planning process aims at finding the open-loop control $\mathbf{u}$ that minimizes the cost function

$$J(\mathbf{u}) = \mathbb{E}\left[q_f(\mathbf{x}_T, T) + \int_0^T q(\mathbf{x}_t, \mathbf{u}(t), t)\, dt\right], \tag{2}$$

where $q$ and $q_f$ are quadratic functions of the state $\mathbf{x}$ and $\mathbb{E}[\cdot]$ denotes the expectation with respect to the random variable $(\boldsymbol{\xi}, \mathbf{x}_0, \mathbf{w})$.

Let us define the mean $\mathbf{m}(t) = \mathbb{E}[\mathbf{x}_t]$, the covariance $\mathbf{P}(t) = \mathbb{E}[(\mathbf{x}_t - \mathbf{m}(t))(\mathbf{x}_t - \mathbf{m}(t))^\top]$ of $\mathbf{x}_t$, and the cross-covariance $\mathbf{D}(t) = \mathbb{E}[(\mathbf{x}_t - \mathbf{m}(t))(\boldsymbol{\xi} - \boldsymbol{\mu})^\top]$ between $\mathbf{x}_t$ and $\boldsymbol{\xi}$. In general the random variables $\mathbf{x}_0$ and $\boldsymbol{\xi}$ are assumed to be uncorrelated so that $\mathbf{D}(0) = \mathbf{0}$. As shown in the Materials and Methods, the uncertain stochastic optimal open-loop control (USOOC) problem defined by Eqs 1 and 2 can be approximated by the following deterministic optimal control problem in augmented state $(\mathbf{m}, \mathbf{P}, \mathbf{D})$.

**Problem 1**. *The problem defined by* Eqs 1 *and* 2 *can be approximated by a deterministic optimal control problem with dynamics*

$$
\begin{cases}
\dot{\mathbf{m}}(t) & = & \mathbf{f}(\mathbf{m}, \mathbf{u}, t; \boldsymbol{\mu}), & \mathbf{m}(0) = \mathbf{m}_0, \\
\dot{\mathbf{P}}(t) & = & \mathbf{A}(t)\mathbf{P}(t) + \mathbf{P}(t)\mathbf{A}(t)^\top + \mathbf{G}(t)\mathbf{G}(t)^\top \\
& & + \mathbf{C}(t)\mathbf{D}(t)^\top + \mathbf{D}(t)\mathbf{C}(t)^\top, & \mathbf{P}(0) = \mathbf{P}_0, \\
\dot{\mathbf{D}}(t) & = & \mathbf{A}(t)\mathbf{D}(t) + \mathbf{C}(t)\boldsymbol{\Sigma}, & \mathbf{D}(0) = \mathbf{0},
\end{cases}
\tag{3}
$$

*where* $\mathbf{A}(t)$, $\mathbf{C}(t)$ *and* $\mathbf{G}(t)$ *are defined from* Eq 1 *as*

$$
\mathbf{A}(t) := \frac{\partial \mathbf{f}}{\partial \mathbf{x}}(\mathbf{m}(t), \mathbf{u}(t), t; \boldsymbol{\mu}), \ \mathbf{C}(t) := \frac{\partial \mathbf{f}}{\partial \boldsymbol{\xi}}(\mathbf{m}(t), \mathbf{u}(t), t; \boldsymbol{\mu}), \ \mathbf{G}(t) := \mathbf{G}(\mathbf{m}(t), \mathbf{u}(t), t),
\tag{4}
$$

*and cost function*

$$
J(\mathbf{u}) = \phi(\mathbf{m}(T), \mathbf{P}(T), T) + \int_0^T \ell(\mathbf{m}(t), \mathbf{P}(t), \mathbf{u}(t), t) \, dt,
\tag{5}
$$

*where the functions* $\phi$ *and* $\ell$ *can be determined from* Eq 2.

To illustrate how the cost function can be obtained, assume for instance that

$$
J(\mathbf{u}) = \mathbb{E}\left[ \mathbf{x}_T^\top \mathbf{Q}_f \mathbf{x}_T + \int_0^T \left( \mathbf{u}(t)^\top \mathbf{R} \mathbf{u}(t) + \mathbf{x}_t^\top \mathbf{Q} \mathbf{x}_t \right) dt \right],
\tag{6}
$$

where $\mathbf{R}$ is positive definite and $\mathbf{Q}$, $\mathbf{Q}_f$ are positive semidefinite matrices of appropriate dimensions. The expectation can then be rewritten only in terms of $\mathbf{m}(t)$ and $\mathbf{P}(t)$ as

$$
J(\mathbf{u}) = \mathbf{m}(T)^\top \mathbf{Q}_f \mathbf{m}(T) + \mathrm{tr}(\mathbf{Q}_f \mathbf{P}(T)) + \int_0^T \left( \mathbf{u}(t)^\top \mathbf{R} \mathbf{u}(t) + \mathbf{m}(t)^\top \mathbf{Q} \mathbf{m}(t) + \mathrm{tr}(\mathbf{Q}\mathbf{P}(t)) \right) dt
\tag{7}
$$

where $\mathrm{tr}(\cdot)$ is the trace operator. We note that a reference trajectory could be added in Eq 6. Also, additional boundary or path constraints could be considered in more general formulations of the problem. The only requirement is to be able to write those additional constraints in terms of the mean and covariance of $\mathbf{x}_t$. For instance, a constraint on the probability of reaching a given target can be added in this framework.

Importantly, the above approach yields exact solutions to the original problem when *(i)* $\mathbf{f}$ is an affine function of $\mathbf{x}$, and *(ii)* $\mathbf{G}$ is independent on $\mathbf{x}$ (see Materials and methods). The case where $\boldsymbol{\xi}$ is deterministic has been treated in details in [22, 23]. There, it was shown that co-contraction is an optimal strategy to minimize effort and variance objectives in presence of internal sensorimotor noise. Here, our focus is instead on the effects of the random variable $\boldsymbol{\xi}$ on the optimal feedforward motor strategy. These effects can be isolated when $\mathbf{G} \equiv \mathbf{0}$ and, therefore, it is interesting to initially consider this scenario. In this case, Eq 1 rewrites as the ODE

$$
\dot{\mathbf{x}}(t) = \mathbf{f}(\mathbf{x}(t), \mathbf{u}(t), t; \boldsymbol{\xi}),
\tag{8}
$$

and the expectations $\mathbb{E}[\cdot]$ are taken with respect to the random variable $(\boldsymbol{\xi}, \mathbf{x}_0)$. If $\mathbf{x}_0$ is perfectly known, then all the uncertainty comes from the environment. To solve this problem, the solution proposed by Problem 1 can be used by setting $\mathbf{G} \equiv \mathbf{0}$. However, an alternative approach consists of extending the original state $\mathbf{x}$ with $s$ copies corresponding to different values of $\boldsymbol{\xi}$. This is readily the case for discrete variables and it can be obtained via discretization when $\boldsymbol{\xi}$ is a continuous variable (e.g., [46–51]). The dimensional advantage of our stochastic linearization approach compared to the discretization one is discussed in the Materials and Methods.

In what follows, we explore how the occurrence of random disturbances coming from the environment affects feedforward control, in particular the planning of mechanical impedance and muscle co-contraction, in a variety of motor control tasks.

## Case of a stabilization task

To illustrate our purpose, we first present a toy example capturing the essence of a stabilization task and allowing us to keep computations analytically tractable. Let us consider the bilinear system

$$\dot{x}(t) = f(t) - k(t)\,x(t) + \xi, \tag{9}$$

where $x$ is the scalar state, $\mathbf{u} = (f, k)$ is the control vector composed of a term $f$ representing a net "force" and a term $k \geq 0$ representing a "stiffness". The parameter $\xi$ represents an external disturbance, which is modeled as a Bernoulli random variable with probability $\alpha$: $\mathrm{pr}(\xi = 1) = \alpha$ and $\mathrm{pr}(\xi = 0) = 1 - \alpha$. Therefore, the mean and variance of $\xi$ are respectively $\mu = \alpha$ and $\Sigma = \alpha(1 - \alpha)$, the latter being a measure of task uncertainty. The random variable $\xi$ represents a disturbance that may occur over the planning time horizon $[0, T]$ with probability $\alpha$ (e.g., as if an external force were to randomly push our hand during repeated attempts to stabilize it).

We thus assume that the optimal control is determined as the one that minimizes the expected quadratic cost with scalar weights $q_f > 0$ and $q \geq 1$,

$$J(\mathbf{u}) = \mathbb{E}\left[ q_f x(T)^2 + \int_0^T \left( f(t)^2 + k(t)^2 + q x(t)^2 \right) dt \right], \tag{10}$$

among the open-loop controls $\mathbf{u}(t)$ for $t \in [0, T]$ ensuring that the expectation $\mathbb{E}[x(T)]$ of the final state equals 0. We assume that the initial state is perfectly known with $x(0) = 0$ (i.e., $m_0 = 0$ and $P_0 = 0$) and that time $T$ is fixed.

Interestingly, it can be proven that when there is some uncertainty in the environment (i.e., $0 < \alpha < 1$), the optimal control satisfies $k > 0$ (see S1 Text for the proof). Reciprocally, when there is no uncertainty in the environment (i.e., $\alpha = 0$ or $\alpha = 1$), the optimal control satisfies $k \equiv 0$. In conclusion, our model predicts a characteristic exploitation of stiffness in response to the very presence of uncertainty in the task.

We performed numerical simulations to visualize the characteristic relationship between uncertainty and stiffness for varying probabilities $\alpha$, with the results shown in Fig 1 (black traces). Note that since the drift is bilinear and the cost is quadratic, our method yields exact solutions in this example. Contrary to the mean optimal net force $f$, which evolves linearly depending on the probability of the external disturbance $\xi$, the evolution of the mean optimal stiffness $k$ is strictly convex upwards and exhibits vertical tangents at $\alpha = 0$ and $\alpha = 1$. Accordingly, the evolution of stiffness with respect to task uncertainty exhibits a logarithmic shape, with a steep increase at low degrees of uncertainty (Fig 1C). Interestingly, if we add a multiplicative noise in these simulations (10% of the input $\mathbf{u}$), the above relationships remain valid, but a lower stiffness becomes optimal in general. Indeed, too large stiffness would increase uncertainty and the expected cost so that it is better to lower the overall stiffness in this case. However, it is worth noting that symmetry is broken with multiplicative noise because the amount of noise scales with $f$. For example, while both $\alpha = 0$ and $\alpha = 1$ do not introduce external uncertainty, the difference lies in the fact that with $\alpha = 1$, we have $f = -1$, which adds internal noise. In contrast, with $\alpha = 0$, $f = 0$, meaning no additional noise occurs. Therefore, it is optimal to slightly increase system stiffness to mitigate this internal motor noise when $\alpha = 1$. Furthermore, if we add some uncertainty about the initial state $x(0)$ (e.g., $P_0 = 0.5$ instead of $P_0 = 0$), the relationship between stiffness and uncertainty remains similar but with an upward

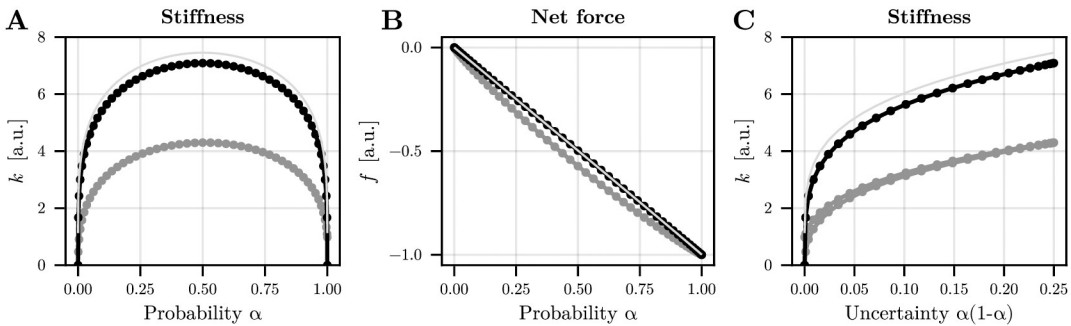

**Fig 1. Optimal feedforward strategy for the toy stabilization task.** A. Evolution of mean stiffness as a function of probability $\alpha$ which determines the occurrence of the external disturbance. Three conditions are depicted: in black, only external noise is considered; in grey, an internal multiplicative motor noise is added as a diffusion term ($0.1k + 0.1f$) to the system (hence becoming an SDE); in plain light grey, the initial state covariance was set to $P_0 = 0.5$ (without multiplicative noise). B. Evolution of mean net force as a function of probability $\alpha$. C. Evolution of mean stiffness as a function of uncertainty $\alpha(1 - \alpha)$, that is, the variance of the external disturbance. Parameters of the simulation were: $T = 5$ s, $q_f = q = 10^4$. Note that the 51 values of $\alpha$ were chosen from the extrema of the Chebyshev polynomial of order 50, to better sample values on the edges of the $[0, 1]$ range.

shift. In such a case, due to the initial state uncertainty, a mean stiffness $k > 1$ is always optimal, even without external uncertainty ($\alpha = 0$ or $\alpha = 1$). Therefore, uncertainty about the initial state tends to increase the mean optimal stiffness.

To further investigate the robustness of this finding and its link with muscle co-contraction, we next simulated an inverted pendulum task with two antagonist muscles, as considered in Hogan's seminal work [3] but with the addition of a random external disturbance. Here, the goal is to maintain the forearm in an upright position for 5 seconds despite the destabilizing action of gravity (see the Methods for more details). In the present simulation, the system has a nonlinear drift due to gravity and is subjected to both internal (additive motor noise) and external uncertainties (Fig 2E). For the external uncertainty, we simulated a random disturbance taking the form of an external torque of 1 Nm applied in each trial with some probability $\alpha$ (Bernoulli variable). We varied $\alpha$ between 0 and 1. The initial state was assumed to be perfectly known such that the case $\alpha = 0$ (no external disturbance) corresponds to the solution depicted in Fig 1D of the reference [23] (i.e., parameters are the same). Interestingly, the characteristic sharp-edged relationship between probability $\alpha$ and stiffness remained evident (Fig 2D). Similar findings were observed for muscle coactivation (Fig 2E). Following [52], we define coactivation as the minimum of the activities of antagonist muscles spanning a joint, and co-contraction as their sum. Hence, co-contraction is proportional to stiffness in our model. As in the toy example, internal uncertainty mainly induces an offset in the latter relationships, meaning that it is optimal to use a nominal level of feedforward stiffness/coactivation, even without environmental uncertainty.

## Case of a reaching task

The two examples above consisted of simple stabilization tasks. We show here that a similar result holds for more complex musculoskeletal systems during reaching tasks (Figs 3 and 4). It is known that when a force field is intermittently applied across trials, participants tend to co-contract muscles (e.g., [38]). Actually, our model suggests that the optimal level of muscle co-contraction should change with the probability of occurrence of the force field ($\alpha$ parameter). To get some insights into how the co-contraction would vary, we simulated a planar reaching task with a two-link arm model actuated by six muscles, as in the Fig 4 of reference [23] (for

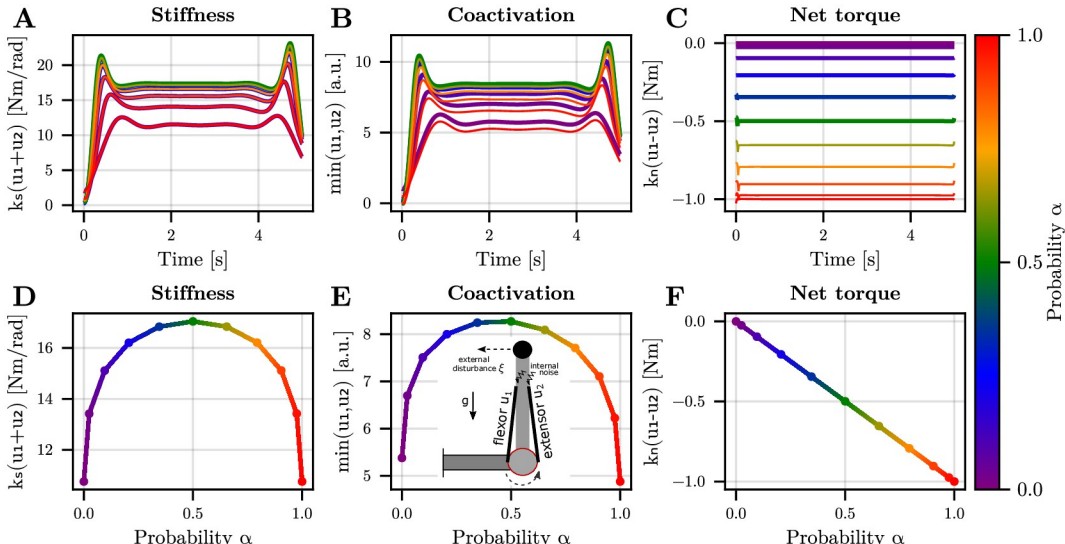

**Fig 2. Optimal feedforward strategy for maintaining the forearm upright.** A. Optimal stiffness trajectory. Stiffness is $k_s(u_1 + u_2)$ where $u_1$ and $u_2$ are respectively the flexor and extensor elbow muscle activations (in arbitrary units). Stiffness is proportional to muscle co-contraction, here defined as $u_1 + u_2$. The probability $\alpha$ is indicated by a color code. Note that the fluctuations surrounding the plateau are due to the initial state (small initial state covariance) and finite-time horizon (allowing specific refinements at the end of the simulation). B. Optimal level of coactivation. Coactivation is here defined as $\min(u_1, u_2)$. C. Optimal net torque $k_n(u_1-u_2)$. D. Evolution of mean stiffness as a function of $\alpha$. E. Evolution of mean coactivation as a function of $\alpha$. The inset illustrates the task and posture. F. Evolution of mean net torque as a function of $\alpha$. Parameters of the simulations were as in [23] (scenario with a load attached to the hand). Note that the 11 values of $\alpha$ were chosen from the extrema of the Chebyshev polynomial of order 10.

more details refer to the Materials and Methods section). The underlying musculoskeletal model is taken from [53]. Here, the task is to reach forward to a target at 25 cm in 750 ms while minimizing the muscle effort, Cartesian acceleration and endpoint variance, subjected to both internal and external uncertainties. The internal uncertainty was additive motor noise acting at the level of joint torques. The external random force field was a velocity-dependent lateral force field (i.e., a force pushing to the right with a magnitude scaling with the forward velocity of the hand), occurring with probability $\alpha$ within the planning horizon. Fig 3 shows that, when $\alpha = 0$ (no force field), a nominal level of stiffness and coactivation is optimal to achieve the task. This means that the intrinsic muscle viscoelasticity is sufficient to deal with internal motor noise affecting task performance. In our setup, co-contraction was negligible during the initial phase of motion, primarily due to the very small initial state covariance used in these simulations. With a larger initial covariance $\mathbf{P}_0$, the simulations would predict a more substantial co-contraction from the start of the motion. When $\alpha = 0.2$, that is when the force field is present in one-fifth of the trials, the optimal solution is to increase activations of muscle pairs, which in turn increases joint stiffness and coactivation (compare left and right columns in Fig 3B, 3C and 3D). For this planar arm reaching task, the evolution of mean stiffness/coactivation (averaged across joints/muscles and over time) as a function of probability and task uncertainty is displayed in Fig 4. It is shown that the steepness of the characteristic relationship is reduced but the pattern remains: the optimal stiffness and coactivation must increase quite steeply as soon as some external uncertainty arises in the task before a gentler increase is observed for larger uncertainties. This is revealed by the logarithmic curve fitting depicted in Fig 4C.

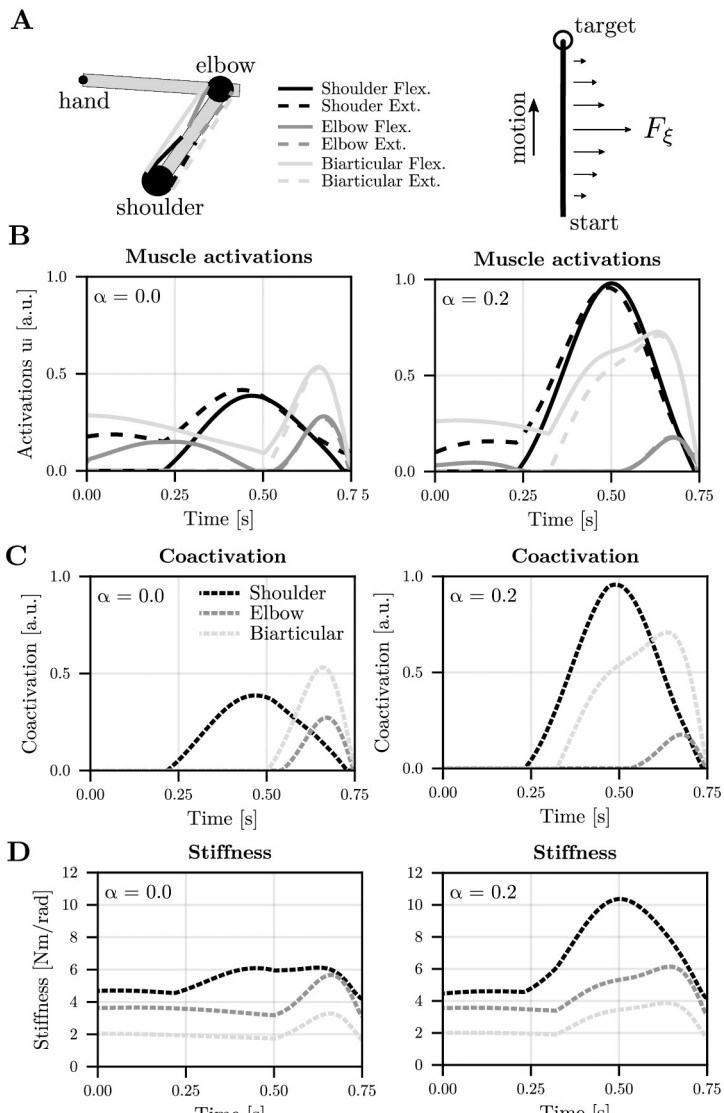

**Fig 3. Optimal feedforward strategy for a planar arm reaching task under uncertainty.** A. Illustration of the 6-muscle arm model, the initial posture and the task with the external disturbance. The horizontal force field is defined as $F_\xi = 1.5\xi\dot{y}$ where $\dot{y}$ is the velocity along the y-axis of the motion. B. Optimal muscle activation pattern (for the 6 muscles) for the $\alpha = 0$ and $\alpha = 0.2$ conditions respectively. In the latter case, $F_\xi$ is triggered with a 20% probability in each trial. C. Optimal coactivation pattern. The 6 muscles were grouped by pairs (1–2, shoulder; 3–4, elbow; 5–6, biarticular). Each pair of muscle was then treated separately (shoulder, elbow and bi-articular) to compute the coactivation with the $\min(u_i, u_{i+1})$ operator, for $i \in \{1, 3, 5\}$. D. Optimal joint stiffness pattern. The values of the diagonal terms (shoulder and elbow) and the off-diagonal (biarticular) term are depicted.

## Experimental testing

To test the characteristic prediction from the computational model on how muscle co-contraction increases with the uncertainty of external disturbances, an experiment involving wrist flexion movements was performed with 16 participants. An active wrist exoskeleton was used to implement random disturbances with different probabilities in 100-trial blocks (Fig 5A and 5B). We designed a protocol such that the mechanical disturbances were applied late in the

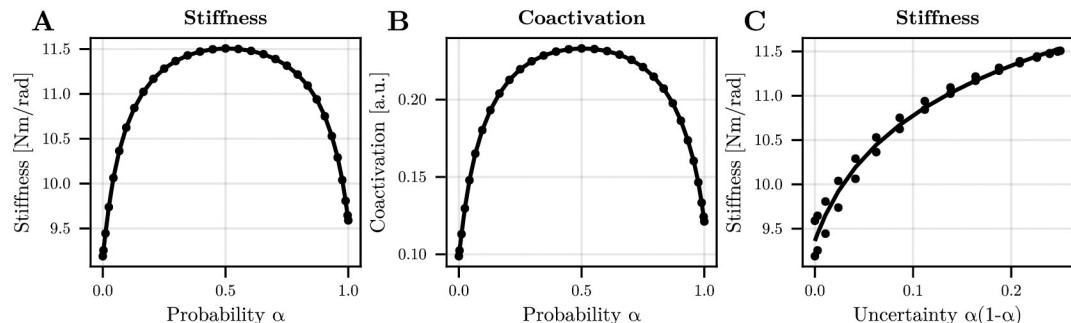

**Fig 4. Evolution of the optimal stiffness and coactivation for the planar arm reaching task under uncertainty.** A. Evolution of mean stiffness as a function of $\alpha$ (computed as the mean value of the trace of the joint stiffness matrix). B. Evolution of mean coactivation as a function of $\alpha$ (computed as the mean of the average coactivation of the 3 agonist-antagonist pairs of muscles). C. Evolution of mean stiffness as a function of external uncertainty $\alpha(1 - \alpha)$. Due to the complex nonlinear dynamics, some hysteresis is observed in these simulations. A logarithm fit is used to describe the relationship between uncertainty and stiffness. Note that the 31 values of $\alpha$ were chosen from the extrema of the Chebyshev polynomial of order 30.

movement to emphasize the role of feedforward control. In this case, due to neural delays, a pure feedback control strategy would be ineffective in reaching the target without overshooting when the disturbance occurs suddenly. Here, the disturbance was a sigmoidal torque plateauing at 0.75 Nm in 500 ms. The random disturbance was a Bernoulli variable of probability $\alpha \in \{0, 0.25, 0.5, 0.75, 1\}$. The two first blocks without uncertainty (i.e., $\alpha = 0$ and $\alpha = 1$) were always performed in this order like in classical motor adaptation protocols. The three other

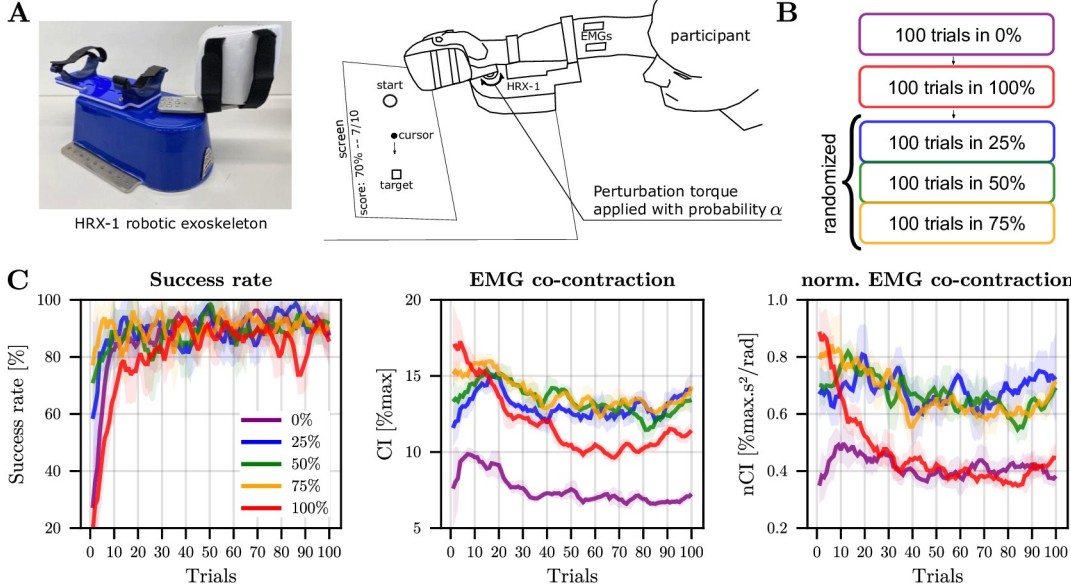

**Fig 5. Task and behavioral adaptation across trials.** A. Picture of the wrist exoskeleton robot and of the apparatus. B. Protocol with the block of trials. C. Adaptation of success rate, EMG co-contraction and normalized EMG co-contraction across blocks of trials. The EMG co-contraction was computed between 150 ms before and 20 ms after the perturbation (based on the trigger position for trials without perturbation). Shaded areas represent the standard deviation across participants. The average evolution of the rate of success over 5 trials, computed over a sliding window for each participant before averaging is depicted. The corresponding trial-by-trial evolution of the EMG co-contraction, averaged across participants, is also depicted as well as the normalized EMG co-contraction.

blocks involving uncertainty were randomized across participants. Details about the experiment are given in the Materials and Methods section. As expected, participants had to adapt their motor strategy to reach the target without undershooting/overshooting in $\sim 500$ ms as required by the protocol (Fig 5C). Differences were observed depending on the disturbance's probability within a block.

The adaptation of the success rate is reported in the first panel of Fig 5C. It reveals that participants quickly learned to perform the task in absence of any perturbation ($\alpha = 0$), with an average success rate reaching a plateau consistently above 80% after a dozen of trials. In the 100% condition, which preceded the $\alpha = \{0.25, 0.50, 0.75\}$ conditions, participants required about 30–50 trials to adapt and consistently achieve a success rate above 80%. After this initial adaptation, participants needed about 10–20 trials to adjust to the uncertain conditions.

We then analyzed EMG co-contraction on the window [-150;20] ms around the disturbance onset, that is, before any neural feedback could influence EMG signals. Fig 5C shows that there was a clear adaptation trend in the $\alpha = 1$ condition, with a plateau attained after about 50 trials for the standard EMG co-contraction index (CI, Eq 27). Given that EMG scales with peak acceleration/deceleration [54], we also considered a normalized EMG co-contraction index (nCI). Indeed, although we attempted to impose a movement time of about 500 ms, participants tended to move slightly faster when the perturbation was present (mean movement times were: $\alpha = 0$%: 575 ms, $\alpha = 25$%: 574 ms, $\alpha = 50$%: 566 ms, $\alpha = 75$%: 562 ms, $\alpha = 100$%: 506 ms). Thus, nCI describes more faithfully the stiffening of human joints as it mitigates the kinematic-dependent effects. About 30 trials were needed to attain a plateau for this normalized EMG co-contraction index (nCI, Eq 28). Interestingly, all conditions appeared to be stabilized in the second half of the block.

To compare adapted motor strategies, as assumed by our optimal control model, we focused on the last 50 trials with nearly constant EMG co-contraction level for the subsequent analyses. The evolution of CI and nCI as a function of the disturbance's probability within a block is reported in Fig 6A and 6B.

After adaptation, we observed higher values of CI when uncertainty was present in the task ($0 < \alpha < 1$). A main effect of the disturbance's probability on CI was found ($F_{4,60} = 17.9$, $p < 10^{-6}$, $\eta^2 = 0.54$). Subsequent post-hoc comparisons showed that CI values were smaller in the 0% than in all other conditions (in all cases: $p < 0.009$, Cohen's $D > 0.99$). Furthermore, coactivation was significantly smaller in the 100% condition than in the 50% and 75% conditions (in both cases: $p < 0.05$, Cohen's $D > 0.72$). Regarding nCI, a main effect of the disturbance's probability was still present ($F_{4,60} = 18.4$, $p < 10^{-6}$, $\eta^2 = 0.55$) and nCI was significantly higher in the 25%, 50% and 75% conditions than in the 0% condition (in all cases: $p < 0.012$, Cohen's $D > 0.94$). Importantly, these three uncertain conditions also exhibited significantly higher nCI than the 100% condition (in all cases: $p < 0.002$, Cohen's $D > 1.2$). In summary, our results confirm that the participants significantly increased EMG co-contraction when environmental uncertainty was added to the task. An important result was the significant decrease of EMG co-contraction in the 100% condition compared to the 75% condition, thus showing that EMG co-contraction does not simply scale with the frequency of the disturbance.

To verify if this trend was consistent with the model's predictions, we simulated this uncertain wrist reaching task (see Materials and methods for details). The predicted evolution of stiffness and co-contraction as a function of the probability $\alpha$ are reported in Fig 6C and 6D. A good match between the experimental EMG co-contraction and the predicted stiffness/co-contraction can be observed, thereby showing the plausibility of theory. Because our protocol limits us to only 3 different values of the variance $\alpha(1 - \alpha)$, we could not display the evolution of uncertainty as a function of stiffness using a logarithmic fit. Furthermore, mechanical

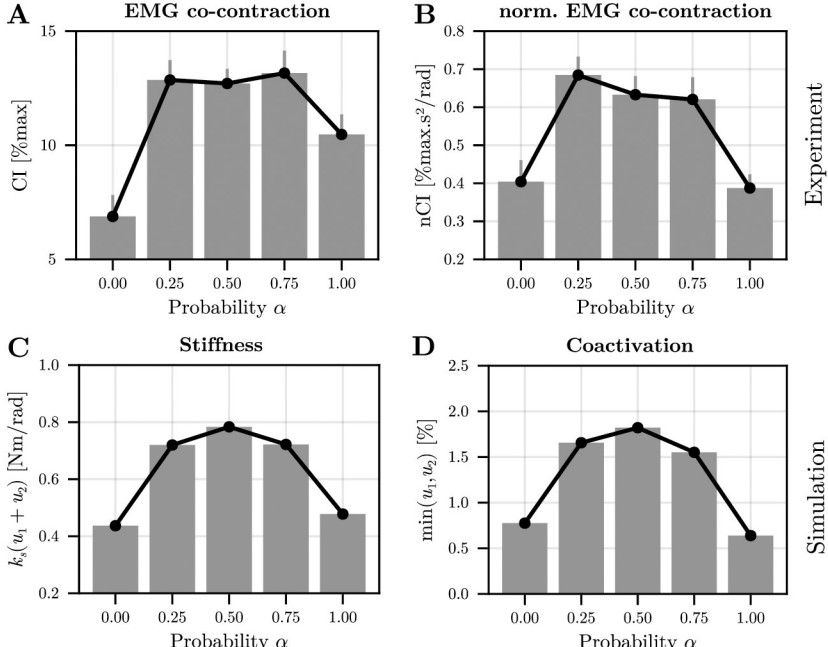

**Fig 6. Changes in EMG co-contraction as a function of the disturbance's probability $\alpha$.** A. EMG co-contraction index (CI) before any neural feedback response could influence EMG signals. Mean values across participants are reported and error bars represent the standard errors of the mean. B. Same information for the normalized EMG co-contraction index (nCI). Evolution of the optimal stiffness as a function of the disturbance's probability. C. Mean stiffness as a function of probability $\alpha$ in simulation. D. Mean co-contraction as a function of probability $\alpha$ in simulation. In simulation, the success rate was about 85%; hence similar to experimental values.

impedance is more than stiffness but damping and stiffness covary in our model, thus we do not display any redundant information about damping.

## Discussion

In the subway, we often hold on to the vertical bar to stabilize our body against the jolty movements of the train. When spreading our legs to widen the base of support is not possible, we can still co-contract our muscles to stiffen the arm and remain steady. The present study investigated this link between environmental uncertainty and impedance control via muscle co-contraction. To this aim, we developed a computational framework that can be applied to nonlinear musculoskeletal systems subjected to internal sensorimotor noise and external random fluctuations. In particular, our modeling focused on the adaptation of the feedforward motor command to the presence of random disturbances induced by the environment. Theoretical considerations and numerical simulations led to the characteristic prediction that muscle co-contraction steeply increases with the uncertainty in the task. To test this prediction, we conducted an experiment involving wrist reaching movements, which confirmed that EMG co-contraction was greater with an uncertain force field compared to conditions where the force field was always turned on or off. Below, we discuss the implications of these findings, the limitations of our modeling and possible extensions.

We have shown that any type of uncertainty, be it internal or external to the CNS, calls for impedance control via muscle co-contraction. Our theoretical and experimental results confirmed that this is part of an optimal feedforward strategy to mitigate the effects of unpredictable

disturbances. The larger muscle co-contraction when facing a greater uncertainty is consistent with a large body of the literature. First, most adaptation studies start with a baseline condition before a specific force field is suddenly turned on. Admittedly, the exposure to a novel force field can be viewed as an increase in task uncertainty. Coherently, muscle co-contraction is generally larger at the beginning of motor adaptation to a novel force field and is progressively reduced through practice, which decreases uncertainty [21, 37]. The same pattern of adaptation was also noticeable in our data for the 100% condition. Second, the study [38] found that co-contraction tends to increase when a force field is randomly turned on and off across trials, which is closely related to the type of uncertainty considered in the present work. Finally, in divergent force fields amplifying the internal noise, participants exploit muscle co-contraction to adjust the endpoint stiffness, even at the end of the adaptation process [4, 6, 37]. However, in this case, it is the very presence of internal sensorimotor noise that makes such external force fields "unpredictable". Indeed, the force field is perfectly deterministic *per se* but, for the participant, it is pushing left or right at random, hence the greater co-contraction. Indeed, simulations without sensorimotor noise do not predict any relevant co-contraction in a divergent force field whereas they do in a random force field. This was verified for the inverted pendulum task where gravity induces a divergent force field on the forearm. This argument illustrates the conceptual difference between the uncertainty originating from the sensorimotor system itself and that from the environment. On the other hand, signal-dependent noise may limit the extent to which co-contraction can reduce kinematic variability (see Fig 1 for instance). Yet, the very origin of signal-dependent noise has been debated [30] and co-contraction tends to decrease kinematic variability in practice [13, 14, 55, 56].

As far as the feedforward component of the motor command is concerned, more uncertainty thus justifies an increase in muscle co-contraction to minimize variability. However, feedback is also present and crucial to human motor control as well as metabolically cheaper than co-contraction in general [15, 45, 57]. To model environmental uncertainty with feedback control, robust H-infinity control is another approach which represents a model-free strategy considering the worst-case scenario in designing the control policy [38, 58, 59]. H-infinity control defines a state-feedback control policy to reject unmodeled disturbances, which leads to larger control gains compared to stochastic optimal control. Nevertheless, H-infinity control does not predict muscle co-contraction and has been restricted to linear systems, making it less applicable to more complex neuromechanical systems. In contrast, our framework can easily accommodate nonlinear systems. Crucially, the approximate deterministic optimal control problem preserves the nonlinearity of the drift's dynamics in the mean propagation, whereas the linearized drift is used only to approximate covariance propagation. However, in our model-based strategy, the mean and variance of the external disturbance must be estimated to optimally adjust the feedforward motor command and the level of co-contraction. This seems to be plausible with respect to the literature [33], and such an inference might be possible through adaptation and learning [60, 61]. Participants may build an internal model of the task, including its uncertainty, based on their prior or recent experience [62]. Concretely here, $\mu$ and $\Sigma$ could be meta-parameters that the brain adapts to modulate co-contraction across repeated trials [21]. After a series of undisturbed trials, it is likely that the brain will decrease $\Sigma$ whereas it will increase it following a disturbance. This could account for the trial-by-trial modulation of co-contraction found in [38]. Therefore, the CNS may integrate knowledge of the task uncertainty to plan subsequent motor commands. More generally, it is likely that a combination of feedforward and feedback control will be used by the CNS to perform uncertain motor tasks [19]. Indeed, feedforward co-contraction likely provides a nominal level of stability in the task, which, in turn, makes feedback control more efficient by allowing larger gains and/or improving muscle reactivity.

It could thus be argued that the absence of feedback-mediated corrections is a weakness in our modeling, and this is a fair point. However, isolating the feedforward component of the motor command was advantageous to unveil the characteristic relationship between uncertainty and muscle co-contraction, and to rigorously set the foundations for future extensions. The main consequence of neglecting the effects of high-level feedback is that the predicted muscle co-contraction or impedance is likely over-estimated in our model. Indeed, sensory feedback would help to perform the task so that a lower co-contraction would probably be required in reality [63]. In the experiment, the disturbance was deliberately introduced at the end of the movement so that the delays in feedback loops were too long to reject the disturbance with feedback-only control. At least two approaches can be envisioned to consider the effects of high-level feedback loops in our framework. The first one is to complement the control scheme with a locally-optimal feedback control law (e.g., linear-quadratic-Gaussian a.k.a. LQG) as in [40, 64]. While the feedback command automatically adapts to the feedforward command in this approach (e.g., the nominal stability offered by co-contraction is taken into account when setting the feedback controller), the reverse is not true. In other words, the presence of feedback does not automatically affect the feedforward control. The second one is to consider a model predictive control (MPC) approach [65–67]. Intermittent feedback control could allow the system to re-estimate its current state and plan a feedforward motor command on a receding horizon [68]. In particular, if the current state estimate has large covariance, our model will immediately command a large stiffness or co-contraction (e.g., see Fig 1). In practice the MPC framework could be interesting to go beyond the trial-by-trial context of in-lab experiments. In that case, the disturbance $\xi$ would just encode the uncertainty of a given disturbance in the subsequent planning horizon. In any case, the cornerstone of high-level feedback control is the state estimation process optimally mixing sensory information and predictions from forward models [69–71]. By definition, unpredictable disturbances cannot be anticipated through forward models and, therefore, only the sensory information (via the innovation) provides relevant cues to update the state estimate after a random disturbance has occurred. Sensory noise and delays may strongly limit the efficiency feedback control in this case. Feedforward co-contraction appears to be one solution employed by the CNS to circumvent this limitation, fully exploiting the variable impedance of muscles and the antagonistic organization of the musculoskeletal system [3].

Although this work focuses on anticipatory co-contraction modeled as a feedforward mechanism, feedback-mediated co-contraction might also play a role [11]. Yet, optimal feedback control typically corrects errors through reciprocal muscle activations; hence, combining USOOC with an LQG layer, as discussed earlier, should result in increased co-contraction alongside reciprocal muscle responses to unexpected disturbances. This would align well with experimental observations showing that voluntary co-contraction is actually accompanied by reciprocal muscle responses to mechanical perturbations [7, 11]. Feedback responses may nonetheless be modulated by co-contraction. As noted earlier, beyond robustifying the system via pre-reflex or built-in responses, co-contraction may enhance the efficiency of high-level feedback responses by increasing reflex gains or reducing muscle response times.

Future work is needed to develop a more comprehensive framework where co-contraction and feedback gains are planned concurrently, accounting for the effects of noise and delays on state estimation and feedback control relevant to the task. Additionally, while this study utilized simple models of force/impedance and muscle viscoelasticity, it could be beneficial to explore more advanced models that accurately represent the intrinsic viscoelastic properties of muscles (e.g., [72, 73]). The proposed framework should be versatile enough to accommodate these complex muscle models, provided their dimensionality remains manageable, and they involve differentiable functions. Further experimental studies with more naturalistic behaviors

would be valuable for testing the theory and deepening our understanding of how humans balance feedforward impedance and feedback control [28]. Overall, this framework has potential applications beyond the modeling of human motor control, including the control of soft robots [74], variable impedance actuators [75], and human-robot interaction for health and manufacturing applications [76].

## Materials and methods

### Ethics statement

The protocols were approved by the Université Paris-Saclay ethical committee for research (CER-Paris-Saclay-2021–048/A1). Written informed consent was obtained from each participant prior to starting the experiments.

### Theoretical developments

Here, we describe the different variants of the uncertain stochastic open-loop optimal control framework, which can be used depending on the task at hand. For clarity in the presentation, we introduce the different variants by order of complexity and start by considering the case without diffusion term $\mathbf{G}$ in Eq 1. We first focus on the affine case, then on the general nonlinear case. We note that the term "affine" is borrowed from the terminology of SDEs: the drift can still imply a nonlinear system from the point of view of control theory.

**Affine case without diffusion.** Let us consider the affine system

$$\dot{\mathbf{x}}(t) = \mathbf{A}(\mathbf{u}(t), t)\mathbf{x}(t) + \mathbf{b}(\mathbf{u}(t), t) + \mathbf{C}(\mathbf{u}(t), t)\boldsymbol{\xi}, \tag{11}$$

where the variables $\mathbf{x}(t) \in \mathbb{R}^n$, $\mathbf{u}(t) \in \mathbb{R}^m$, $\boldsymbol{\xi} \in \mathbb{R}^p$ are defined as in the Results section. The terms $\mathbf{A}$, $\mathbf{b}$ and $\mathbf{C}$ are assumed to be known functions of the time $t \in [0, T]$ and of the open-loop control $\mathbf{u}(t)$. This dependency on $\mathbf{u}$ can be used to consider bilinear drifts, corresponding to the simplest mechanical model for obtaining meaningful predictions about the planning of mechanical impedance (e.g., as illustrated in the toy example). In practice, these terms represent the internal model of the limb's dynamics and of the task, built by the CNS through learning. The cost function to minimize is in the form of Eq 2. The problem is to find the open-loop control $\mathbf{u}(t)$ and the trajectory distribution $\mathbf{x}(t)$ starting from an initial state $\mathbf{x}(0) = \mathbf{x}_0$ (which can be a random variable with mean $\mathbf{m}_0$ and covariance $\mathbf{P}_0$) minimizing the chosen expected cost.

To solve this problem, we will formulate an equivalent deterministic optimal control problem. To do so, let us express the ODE satisfied by the mean $\mathbf{m}(t)$ and covariance $\mathbf{P}(t)$ of $\mathbf{x}$.

A direct computation shows that the propagation of the mean $\mathbf{m}(t) = \mathbb{E}[\mathbf{x}(t)]$ is given by the ODE

$$\dot{\mathbf{m}}(t) = \mathbf{A}(\mathbf{u}(t), t)\mathbf{m}(t) + \mathbf{b}(\mathbf{u}(t), t) + \mathbf{C}(\mathbf{u}(t), t)\boldsymbol{\mu}, \qquad \mathbf{m}(0) = \mathbf{m}_0, \boldsymbol{\mu} = \mathbb{E}[\boldsymbol{\xi}]. \tag{12}$$

In order to get an equation for the propagation of the $n \times n$ covariance matrix $\mathbf{P}(t) = \mathbb{E}[(\mathbf{x}(t) - \mathbf{m}(t))(\mathbf{x}(t) - \mathbf{m}(t))^\top]$, we introduce the $n \times p$ cross-covariance matrix $\mathbf{D}(t) = \mathbb{E}[(\mathbf{x}(t) - \mathbf{m}(t))(\boldsymbol{\xi} - \boldsymbol{\mu})^\top]$ between $\mathbf{x}$ and $\boldsymbol{\xi}$, and the $p \times p$ covariance $\boldsymbol{\Sigma} = \mathbb{E}[(\boldsymbol{\xi} - \boldsymbol{\mu})(\boldsymbol{\xi} - \boldsymbol{\mu})^\top]$ of $\boldsymbol{\xi}$. Again, a direct computation shows that the covariance propagation is governed by the ODE

$$\begin{cases} \dot{\mathbf{P}}(t) &= \mathbf{A}(\mathbf{u}(t), t)\mathbf{P}(t) + \mathbf{P}(t)\mathbf{A}(\mathbf{u}(t), t)^\top \\ &\quad + \mathbf{C}(\mathbf{u}(t), t)\mathbf{D}(t)^\top + \mathbf{D}(t)\mathbf{C}(\mathbf{u}(t), t)^\top, \\ \dot{\mathbf{D}}(t) &= \mathbf{A}(\mathbf{u}(t), t)\mathbf{D}(t) + \mathbf{C}(\mathbf{u}(t), t)\boldsymbol{\Sigma}. \end{cases} \tag{13}$$

The initial conditions of $\mathbf{P}$ and $\mathbf{D}$ are respectively the covariance of the initial state $\mathbf{P}(0) = \mathbb{E}[(\mathbf{x}_0 - \mathbf{m}_0)(\mathbf{x}_0 - \mathbf{m}_0)^\top] = \mathbf{P}_0$ and $\mathbf{D}(0) = \mathbb{E}[(\mathbf{x}_0 - \mathbf{m}_0)(\boldsymbol{\xi} - \boldsymbol{\mu})^\top]$. As already mentioned, the random variables $\mathbf{x}_0$ and $\boldsymbol{\xi}$ are generally uncorrelated so that we assume $\mathbf{D}(0) = \mathbf{0}$.

**Problem 2**. *For the affine case, the original uncertain optimal open-loop control problem can be replaced by an equivalent deterministic optimal control problem in augmented state* $(\mathbf{m}, \mathbf{P}, \mathbf{D})$ *with dynamics*

$$
\begin{cases}
\dot{\mathbf{m}}(t) = \mathbf{A}(\mathbf{u}(t),t)\mathbf{m}(t) + \mathbf{b}(\mathbf{u}(t),t) + \mathbf{C}(\mathbf{u}(t),t)\boldsymbol{\mu}, & \mathbf{m}(0) = \mathbf{m}_0, \\
\dot{\mathbf{P}}(t) = \mathbf{A}(\mathbf{u}(t),t)\mathbf{P}(t) + \mathbf{P}(t)\mathbf{A}(\mathbf{u}(t),t)^\top \\
\qquad\quad + \mathbf{C}(\mathbf{u}(t),t)\mathbf{D}(t)^\top + \mathbf{D}(t)\mathbf{C}(\mathbf{u}(t),t)^\top, & \mathbf{P}(0) = \mathbf{P}_0, \\
\dot{\mathbf{D}}(t) = \mathbf{A}(\mathbf{u}(t),t)\mathbf{D}(t) + \mathbf{C}(\mathbf{u}(t),t)\boldsymbol{\Sigma}, & \mathbf{D}(0) = \mathbf{0},
\end{cases}
\tag{14}
$$

*and cost function*

$$
J(\mathbf{u}) = \phi(\mathbf{m}(T), \mathbf{P}(T), T) + \int_0^T \ell(\mathbf{m}(t), \mathbf{P}(t), \mathbf{u}(t), t)\, dt. \tag{15}
$$

From the above, it can be noted that we made no assumption on the distribution of the random variable $\boldsymbol{\xi}$, and that the optimal solution only depends on the mean $\boldsymbol{\mu}$ and covariance $\boldsymbol{\Sigma}$ of $\boldsymbol{\xi}$. Furthermore, additional boundary conditions that would only involve the mean and covariance (and possibly the cross-covariance as well) can be easily considered.

If the initial state $\mathbf{x}_0$ is deterministic (i.e., $\mathbf{x}_0 = \mathbf{m}_0$), the problem admits a much simpler expression. Because $\mathbf{P}(0) = \mathbf{0}$ and $\mathbf{D}(0) = \mathbf{0}$, we can express $\mathbf{P}(t)$ as a function of $\mathbf{D}(t)$ as follows:

$$
\mathbf{P}(t) = \mathbf{D}(t)\boldsymbol{\Sigma}^{-1}\mathbf{D}(t)^\top. \tag{16}
$$

The equivalent deterministic optimal control problem drastically simplifies as it can be fully written in terms of the augmented state $(\mathbf{m}, \mathbf{D})$, and $\mathbf{P}$ can be directly computed from Eq 16.

**General nonlinear case without diffusion.** Let us consider the same problem but with an uncertain nonlinear system of the form

$$
\dot{\mathbf{x}}(t) = \mathbf{f}(\mathbf{x}(t), \mathbf{u}(t), t; \boldsymbol{\xi}). \tag{17}
$$

In this case, it is necessary to resort to approximation techniques. Again, we introduce the mean $\mathbf{m}_{\mathbf{x}} = \mathbb{E}[\mathbf{x}]$ and covariance $\mathbf{P}_{\mathbf{x}} = \mathbb{E}[(\mathbf{x} - \mathbf{m}_{\mathbf{x}})(\mathbf{x} - \mathbf{m}_{\mathbf{x}})^\top]$ of $\mathbf{x}$. The subscript $\mathbf{x}$ is introduced here because, in contrast to the affine case, now $\mathbf{m}_{\mathbf{x}}(t)$ and $\mathbf{P}_{\mathbf{x}}(t)$ cannot be obtained as solutions of a finite-dimensional control system. Indeed, the nonlinearity of $\mathbf{f}$ may introduce moments of any order. However, using statistical linearization techniques (see [22, 77]), we can approximate these quantities by $(\mathbf{m}, \mathbf{P})(t)$, the first two components of the trajectories $(\mathbf{m}, \mathbf{P}, \mathbf{D})(t)$ of

$$
\begin{cases}
\dot{\mathbf{m}}(t) = \mathbf{f}(\mathbf{m}(t), \mathbf{u}(t), t; \boldsymbol{\mu}), & \mathbf{m}(0) = \mathbf{m}_0, \\
\dot{\mathbf{P}}(t) = \mathbf{A}(t)\mathbf{P}(t) + \mathbf{P}(t)\mathbf{A}(t)^\top + \mathbf{C}(t)\mathbf{D}(t)^\top + \mathbf{D}(t)\mathbf{C}(t)^\top, & \mathbf{P}(0) = \mathbf{P}_0, \\
\dot{\mathbf{D}}(t) = \mathbf{A}(t)\mathbf{D}(t) + \mathbf{C}(t)\boldsymbol{\Sigma}, & \mathbf{D}(0) = \mathbf{0},
\end{cases}
\tag{18}
$$

where

$$\mathbf{A}(t) := \frac{\partial \mathbf{f}}{\partial \mathbf{x}}(\mathbf{m}(t), \mathbf{u}(t), t; \boldsymbol{\mu}), \quad \mathbf{C}(t) := \frac{\partial \mathbf{f}}{\partial \boldsymbol{\xi}}(\mathbf{m}(t), \mathbf{u}(t), t; \boldsymbol{\mu}),$$

$$\boldsymbol{\mu} = \mathbb{E}[\boldsymbol{\xi}], \qquad\qquad \boldsymbol{\Sigma} = \mathbb{E}[(\boldsymbol{\xi} - \boldsymbol{\mu})(\boldsymbol{\xi} - \boldsymbol{\mu})^{\top}].$$

(19)

The latter system can be obtained by augmenting the system's state with $\boldsymbol{\xi}$ and including all the uncertainty in the initial state, and then use the method in [22] with a null diffusion term (see also the proof of Lemma 1 in the S1 Text). Alternatively, we can directly write the propagation of $\mathbf{m_x}(t)$ and $\mathbf{P_x}(t)$ as in Eq. 14 of the S1 Text and use a first order Taylor series expansion to obtain the above equations.

*Remark.* Although referred to as statistical linearization, it is worth noting that the propagation of the mean retains the nonlinear characteristics of the drift (i.e., the deterministic dynamics). The linearized drift around the mean trajectory is primarily used to approximate the propagation of the covariance. This approximation is generally valid if the actual trajectories do not significantly deviate from the mean. In practice, this aligns with the goal of open-loop control (and, more broadly, feedback control) to reduce variance, thereby justifying the approximation. For a more rigorous treatment of these approximations, see [78].

**Problem 3**. *For the general nonlinear case, the original uncertain optimal open-loop control problem can be approximated by a deterministic optimal control problem in augmented state* ($\mathbf{m}$, $\mathbf{P}$, $\mathbf{D}$) *among the trajectories* ($\mathbf{m}$, $\mathbf{P}$, $\mathbf{D}$)$(t)$ *satisfying* Eq 18 *and minimizing the cost given in* Eq 15.

*Remark.* In general, the dimension of the augmented state will be $n\left(p + 1 + \frac{n+1}{2}\right)$. Therefore, it grows quadratically with the dimension of the state $\mathbf{x}$ but only linearly with the dimension of the random parameter $\boldsymbol{\xi}$. Interestingly, if there is no uncertainty about the initial state $\mathbf{x}_0$, the dimension of the augmented state reduces to $n(p + 1)$. Alternative approaches propose to directly discretize the parameter space $\boldsymbol{\xi}$ [46–51], the expected cost being then approximated by a weighted finite sum. If $s$ samples of $\boldsymbol{\xi}$ are used, an associated deterministic optimal control problem in dimension $ns$ can be formulated using $s$ copies of the dynamical system. We refer the reader to the S1 Text where this kind of approach is explicitly developed for a discrete random variable $\boldsymbol{\xi}$. The challenge when such approaches are applied to continuous random variables is to find a small number $s$ allowing accurate approximations of the expected cost. Comparing dimensions, we see that our method might be advantageous if $2s > 2p + n + 3$ or $s > p + 1$ if $\mathbf{x}_0$ is deterministic. If we consider the internal uncertainty coming from the diffusion term, the covariance must be added anyway in the augmented state (see S1 Text), and the approach with $s$ copies requires to solve a deterministic problem in $n\left(\frac{n+1}{2}\right)s$ dimensions, such that our approach can become even more advantageous when $s$ is too large.

In a practical setting (compactly supported dynamics), the accuracy of the approximation in Problem 3 we make is guaranteed by the following result (see S1 Text for the proof and [78] for more general results of this kind).

**Proposition 1**. *Assume that* $\mathbf{f}$ *is smooth, that the set U of control values is a compact subset of* $\mathbb{R}^m$, *that the random vector* $\boldsymbol{\xi}$ *takes values in a compact set, and that all the vector fields* $\mathbf{x} \mapsto \mathbf{f}(\mathbf{x}, \mathbf{u}; \boldsymbol{\xi})$ *have the same compact support. Let* $T > 0$. *Then* ($\mathbf{m}$, $\mathbf{P}$)$(\cdot)$ *converges uniformly on* $[0, T]$ *to* ($\mathbf{m_x}$, $\mathbf{P_x}$)$(\cdot)$ *when* ($\mathbf{P}(\cdot)$, $\mathbf{D}(\cdot)$, $\boldsymbol{\Sigma}$) *converges uniformly to* 0.

*Moreover, if the dynamics is affine with respect to the random parameter* $\boldsymbol{\xi}$, *then the convergence holds independently on* $\boldsymbol{\Sigma}$, *that is, when* ($\mathbf{P}(\cdot)$, $\mathbf{D}(\cdot)$) *converges uniformly to* 0.

**Affine case with diffusion.** Let us now consider the case where the dynamical system is an affine SDE with respect to the state and the uncertain parameter, that is,

$$d\mathbf{x}_t = (\mathbf{A}(\mathbf{u}(t), t)\mathbf{x}_t + \mathbf{b}(\mathbf{u}(t), t) + \mathbf{C}(\mathbf{u}(t), t)\boldsymbol{\xi})dt + \mathbf{G}(\mathbf{u}(t), t)\, d\mathbf{w}_t.$$

(20)

With the diffusion term, the problem is more complicated but can still be solved by formulating an equivalent deterministic optimal control problem with augmented state.

To derive the result, first remind that the cost $J(\mathbf{u})$ in Eq 2 can be rewritten as a function of the mean $\mathbf{m}_\mathbf{x} = \mathbb{E}[\mathbf{x}]$ and covariance $\mathbf{P}_\mathbf{x} = \mathbb{E}[(\mathbf{x} - \mathbf{m}_\mathbf{x})(\mathbf{x} - \mathbf{m}_\mathbf{x})^\top]$ of the process $\mathbf{x}_t$, where the expectation is taken with respect to $\boldsymbol{\xi}$, $\mathbf{x}_0$ and $\mathbf{w}$. Second, it is convenient to distinguish the role of the two sources of uncertainty in the following way. For a fixed instance of the parameter $\boldsymbol{\xi}$, let $\mathbf{x}^{\boldsymbol{\xi}}$ be the solution of Eq 1 with initial condition $\mathbf{x}_0$ and let $\mathbf{m}_{\mathbf{x}^\xi}$, $\mathbf{P}_{\mathbf{x}^\xi}$ be respectively its mean and covariance with respect to $(\mathbf{x}_0, \mathbf{w})$. Then, denoting by $\mathbb{E}_\xi$ the expectation with respect to $\boldsymbol{\xi}$, a computation shows that

$$\mathbf{m}_\mathbf{x} = \mathbb{E}_\xi[\mathbf{m}_{\mathbf{x}^\xi}], \qquad \mathbf{P}_\mathbf{x} = \mathbf{P}_1 + \mathbf{P}_2, \tag{21}$$

where

$$\mathbf{P}_1 = \mathbb{E}_\xi[\mathbf{P}_{\mathbf{x}^\xi}], \quad \mathbf{P}_2 = \mathbb{E}_\xi[(\mathbf{m}_{\mathbf{x}^\xi} - \mathbf{m}_\mathbf{x})(\mathbf{m}_{\mathbf{x}^\xi} - \mathbf{m}_\mathbf{x})^\top]. \tag{22}$$

The covariance $\mathbf{P}_{\mathbf{x}^\xi}$ satisfies the differential equation $\dot{\mathbf{P}}_{\mathbf{x}^\xi} = \mathbf{A}(\mathbf{u})\mathbf{P}_{\mathbf{x}^\xi} + \mathbf{P}_{\mathbf{x}^\xi}\mathbf{A}(\mathbf{u})^\top + \mathbf{G}(\mathbf{u})\mathbf{G}(\mathbf{u})^\top$ with $\mathbf{P}_0 = \mathbb{E}[(\mathbf{x}_0 - \mathbf{m}_0)(\mathbf{x}_0 - \mathbf{m}_0)^\top]$ as an initial condition. Neither the differential equation nor the initial condition depend on $\boldsymbol{\xi}$, so $\mathbf{P}_{\mathbf{x}^\xi}$ does not depend on $\boldsymbol{\xi}$, which implies that $\mathbf{P}_1$ is obtained as the solution of

$$\dot{\mathbf{P}}_1(t) = \mathbf{A}(\mathbf{u}(t),t)\mathbf{P}_1(t) + \mathbf{P}_1(t)\mathbf{A}(\mathbf{u}(t),t)^\top + \mathbf{G}(\mathbf{u}(t),t)\mathbf{G}(\mathbf{u}(t),t)^\top, \quad \mathbf{P}_1(0) = \mathbf{P}_0. \tag{23}$$

On the other hand, $\mathbf{m}_{\mathbf{x}^\xi}$ is solution of the ODE

$$\dot{\mathbf{m}}_{\mathbf{x}^\xi} = \mathbf{A}(\mathbf{u}(t),t)\mathbf{m}_{\mathbf{x}^\xi} + \mathbf{b}(\mathbf{u}(t),t) + \mathbf{C}(\mathbf{u}(t),t)\boldsymbol{\xi}, \qquad \mathbf{m}_{\mathbf{x}^\xi}(0) = \mathbb{E}[\mathbf{x}_0] = \mathbf{m}_0.$$

Arguing as in the affine case without diffusion, we obtain that $(\mathbf{m}_\mathbf{x}, \mathbf{P}_2)$ are the first two components of the trajectories $(\mathbf{m}_\mathbf{x}, \mathbf{P}_2, \mathbf{D})(t)$ of

$$\begin{cases} \dot{\mathbf{m}}_\mathbf{x}(t) &= \mathbf{A}(\mathbf{u}(t),t)\mathbf{m}_\mathbf{x}(t) + \mathbf{b}(\mathbf{u}(t),t) + \mathbf{C}(\mathbf{u}(t),t)\boldsymbol{\mu}, \quad \mathbf{m}_\mathbf{x}(0) = \mathbf{m}_0, \\[4pt] \dot{\mathbf{P}}_2(t) &= \mathbf{A}(\mathbf{u}(t),t)\mathbf{P}_2(t) + \mathbf{P}_2(t)\mathbf{A}(\mathbf{u}(t),t)^\top \\[2pt] & \quad + \mathbf{C}(\mathbf{u}(t),t)\mathbf{D}(t)^\top + \mathbf{D}(t)\mathbf{C}(\mathbf{u}(t),t)^\top, \qquad \mathbf{P}_2(0) = \mathbf{0}, \\[4pt] \dot{\mathbf{D}}(t) &= \mathbf{A}(\mathbf{u}(t),t)\mathbf{D}(t) + \mathbf{C}(\mathbf{u}(t),t)\boldsymbol{\Sigma}, \qquad \mathbf{D}(0) = \mathbf{0}, \end{cases} \tag{24}$$

where $\boldsymbol{\mu}$ and $\boldsymbol{\Sigma}$ are respectively the mean and covariance of $\boldsymbol{\xi}$.

**Problem 4**. *For the affine case with a diffusion term, the original uncertain stochastic optimal open-loop control problem is equivalent to the following deterministic one: minimize the cost $J(\mathbf{u})$ written as a function of $\mathbf{m}_\mathbf{x}$ and $\mathbf{P}_\mathbf{x} = \mathbf{P}_1 + \mathbf{P}_2$ as in Eq 15 among the trajectories $(\mathbf{m}_\mathbf{x}, \mathbf{P}_1, \mathbf{P}_2, \mathbf{D})(t)$ of Eqs (23) and (24).*

**General nonlinear case with diffusion.** For the general nonlinear case where the system writes

$$d\mathbf{x}_t = \mathbf{f}(\mathbf{x}_t, \mathbf{u}(t), t; \boldsymbol{\xi})\, dt + \mathbf{G}(\mathbf{x}_t, \mathbf{u}(t), t)\, d\mathbf{w}_t, \tag{25}$$

we can combine the results of [22] and of the nonlinear case without diffusion described above to propose a deterministic optimal problem approximating the original one. This deterministic problem is constructed as follows: its cost is obtained by replacing in Eq 15 the mean $\mathbf{m}_\mathbf{x}$ and covariance $\mathbf{P}_\mathbf{x}$ by $\mathbf{m}$ and $\mathbf{P} = \mathbf{P}_1 + \mathbf{P}_2$ respectively, and $(\mathbf{m}, \mathbf{P}, \mathbf{D})(t)$ are obtained by combining Eqs 18, 23 and 24 to obtain the USOOC problem approximation given in Problem 1.

### Uncertain wrist reaching experiment

**Experimental protocol.**   A total of $N = 16$ participants were recruited for the experiment to test the model's prediction. They were healthy right-handed adults with the following characteristics: age 27 ± 5 years old, height 177 ± 7.1 cm, weight 73 ± 9.0 kg, hand length 19.7 ± 1.0 cm. The hand length was measured for each participant between the tip of the middle finger and the wrist centre (estimated as the middle of the segment formed by the head of the ulna and the radial styloid process). This study used the HRX-1 (HumanRobotix, London, UK) wrist exoskeleton to record the motion and implement the mechanical disturbance. This device is a 1-degree-of-freedom active wrist exoskeleton controlled at 100 Hz. It offers high backdriveability with low friction and minimal apparent inertia, through the direct transmission of torque between the user and the motor axis, without the interference of any gears. It is equipped with an encoder (Encoder MILE 512–6400, 6400 counts per turn) to measure the position of the human wrist. The participant's hand and forearm were each attached to the exoskeleton with two cuffs. Their respective positions were adjusted closer or further from the exoskeleton's joint axis to match the participant's forearm length.

The task involved 60˚ wrist flexion and extension reaching movements centered around 0˚, which corresponds to the hand aligned with the forearm. Participants were asked to reach targets that were displayed on a screen as 3-cm long green squares. Two targets were considered: left (wrist flexion posture) and right (wrist extension posture). For both targets, participants had to remain static and inside the target during 1.5 s to validate the trial. Whenever a target was validated, it disappeared, and the other target appeared. The right target was always green whereas the left target was initially blue for 500 ms, then green for 500 ms and then blue again until target validation. For a successful trial, the participant had to attain the left target during the green period but could start moving during the first blue period, yielding valid movements for durations between 0.5 s and 1 s when considering both reaction time and motion time. If the target was attained before or after the green period, the trial was failed and if an overshoot was detected, the left target turned red, and the trial was failed.

During the flexion movements, a mechanical disturbance was applied in the direction of the movement at 90% of its full amplitude, triggered with a certain probability across a series of trials. Five different probabilities of disturbance were tested across 5 separate blocks of 100 trials. Hence 500 flexion movements per participant were recorded and analyzed. The disturbance was presented with a Bernoulli distributed probability $\alpha$. In the first block, no disturbance was applied ($\alpha = 0$), so that participants could familiarize themselves with the task and perform it seamlessly. In the second block, the disturbance was always present ($\alpha = 1$) so that participants could adapt to it and potentially learn to compensate for it. The three next blocks were randomized and the disturbance probability in these blocks was one of the following: $\alpha \in \{0.25, 0.5, 0.75\}$. To limit muscle fatigue, 3-minute breaks were imposed between blocks. To motivate participants, their instantaneous rate of success during the block was displayed (both absolute and percentage) and they were told a fictitious highest success rate obtained by the "best participant" so far (selected between 90% and 95%).

The disturbance applied by the HRX-1 robot to the human wrist took the form of a sigmoidal torque with a plateau at $\tau_{\max} = 0.75$ Nm that was reached in roughly 500 ms. When the 90% threshold was crossed, the disturbance torque was generated as follows:

$$\tau_e(t) = \frac{\tau_{\max}}{\left(1 + 2e^{-\gamma(t-\sigma)}\right)^2} \tag{26}$$

where $\gamma = 9.9903$ and $\sigma = 0.0652$. These values were chosen to ensure 500 ms between 5% and 95% of the plateau. Note that the disturbance plateau was maintained longer than the time

necessary to validate a trial (i.e., 2.5 s). Thus, the release of the disturbance could not impact the performance in the task.

To assess muscle co-contraction, the activity of the flexor carpi radialis (wrist flexor) and the extensor carpi radialis (wrist extensor) were recorded using bipolar surface EMG (Wave Plus, Wireless EMG, sample rate 2 kHz; Cometa, Bareggio, Italy). The EMGs were placed according to the SENIAM recommendations [79]. Before placing the electrodes, the skin was locally shaved and cleaned with a hydro-alcoholic solution.

**Data processing.** EMG signals were band-passed filtered (Butterworth, $4^{th}$ order, [20; 450] Hz cut-off frequencies), centered and rectified [80]. They were then normalized by the maximum value obtained for each muscle over the course of the whole experiment. The averaged sum of the two signals over different time windows of interest were used as an EMG co-contraction index, defined as

$$\text{CI} = \frac{1}{0.17} \int_{t_0 - 0.15}^{t_0 + 0.02} \left( u_F(t) + u_E(t) \right) dt \tag{27}$$

where $t_0$ is the onset time of the disturbance, and $u_F(t)$ and $u_E(t)$ are processed EMG signals of the flexor and extensor respectively. Note that this time window was chosen such that only anticipatory and pre-reflex EMG activity was analyzed (hence it excludes any reflex occurring after the mechanical perturbation). The term EMG co-contraction is used here because of the additive nature of the computed index.

Given that muscle activity is known to correlate with joint acceleration/deceleration for single-joint movements [54], we also computed a normalized EMG co-contraction index that accounts for motion deceleration as follows:

$$\text{nCI} = \frac{\text{CI}}{\text{PD}}, \tag{28}$$

where PD is the peak of deceleration of the considered reaching movement. This normalization was chosen because the disturbance was applied near the end of the movement, that is, during the deceleration phase.

Movements were segmented based on motion kinematics as described below. Wrist joint angles were measured at 100 Hz with the encoder of the HRX-1 exoskeleton. Successive positions of the wrist were low-pass filtered (Butterworth, $5^{th}$ order, 5 Hz cut-off frequency). Wrist joint angular velocity and acceleration were obtained through numerical differentiation. Individual movements were first isolated based on the time spent by participants inside targets. Then, for each movement, initial and final times were computed using a threshold at 5% of the peak wrist angular velocity. The kinematics, task events (i.e. targets appearing and disappearing) and muscle activities were all synchronously collected using a Matlab (R2023b, Mathworks, USA) custom code.

**Statistical analyses.** Main effects of the level of uncertainty were first assessed using one-way repeated measurements ANOVA. In case sphericity conditions were not satisfied (i.e. $\epsilon < 0.75$), a Greenhouse-Geisser correction was applied. For all significant ANOVA, we report the $\eta^2$ as a measure of the effect size. The significance level of ANOVA was set at $p < 0.05$. In case a main effect was found, we performed pairwise $t$-tests between the different levels of uncertainty. For all significant comparisons, we report Cohen's D as a measure of the effect size. The level of significance of post-hoc comparisons was set at $p < 0.05$. All statistical analyses were performed using custom Python 3.8 scripts and the Pingouin package [81].

## Modeling of the uncertain 1-dof postural task

For the inverted pendulum task with the forearm, the state vector is $\mathbf{x} = (\theta, \dot{\theta})$ where the elements are respectively the joint angle and velocity, and the control vector is $\mathbf{u} = (u_1, u_2)$ where $u_1$ is the flexor muscle activation and $u_2$ is the extensor muscle activation (in arbitrary units). Note that we will skip certain time dependencies for the sake of readability.

The drift was defined as:

$$\mathbf{f}(\mathbf{x}, \mathbf{u}, t; \boldsymbol{\xi}) = \begin{bmatrix} \dot{\theta} \\ (k_n(u_1 - u_2) - k_s(u_1 + u_2)\theta + k_g \sin(\theta) - b\dot{\theta} + \xi)/I \end{bmatrix} \tag{29}$$

where $k_n = 1Nm$ and $k_s = 1Nm/rad$ are constants, $k_g = 10.754Nm$ is from gravity torque (corresponding to the forearm with a load of 2.268 kg attached at the hand), $b = 1Nms/rad$ is a damping factor, $I = 0.337kg.m^2$ is the moment of inertia with the load, and $\xi$ is the external random disturbance ($\xi = 1Nm$ with probability $\alpha$). Note that the drift is nonlinear because of the sine function and the product terms between $\mathbf{u}$ and $\mathbf{x}$. From the above, $u_1 - u_2$ allows to control the net torque whereas $u_1 + u_2$ allows to control the stiffness.

The diffusion term was defined as:

$$\mathbf{G}(\mathbf{x}, \mathbf{u}, t) = \begin{bmatrix} 0 \\ \sigma_a \end{bmatrix} \tag{30}$$

where $\sigma_a = 0.1$ is the additive noise coefficient and the associated Wiener process was scalar.

The cost function was defined as:

$$J(\mathbf{u}) = \mathbb{E}\left[ \mathbf{x}_T^\top \mathbf{Q}_f \mathbf{x}_T + \int_0^T (\mathbf{u}(t)^\top \mathbf{R}\mathbf{u}(t) + \mathbf{x}_t^\top \mathbf{Q}\mathbf{x}_t)\, dt \right] \tag{31}$$

where $\mathbf{R} = \mathbf{I}$ where $\mathbf{I}$ is the 2x2 identity matrix, $\mathbf{Q} = \mathbf{Q}_f = \text{diag}(10^4, 10^3)$ and $T = 5\ s$. The initial state $\mathbf{x}_0$ had zero mean and covariance $10^{-5}\mathbf{I}$. This small covariance was chosen such that the initial state can be considered as perfectly known but the Cholesky decomposition can still be computed (see below).

Without the term $\xi$, this simulation is the same as the one presented in [23] with the additional load.

## Modeling of the uncertain 2-dof reaching task

For the planar arm reaching task, the equations are more involved but they follow [53] and [23]. The main difference here is that we consider an additional random external disturbance.

In this case, the state vector is $\mathbf{x} = (\theta_1, \theta_2, \dot{\theta}_1, \dot{\theta}_2)$ where the elements are respectively the joint angle and velocity of the shoulder and elbow. The control vector is $\mathbf{u} = (u_1, u_2, u_3, u_4, u_5, u_6)$, which corresponds to all 6 muscle inputs (given in arbitrary units). Again, note that we skip certain time dependencies for the sake of readability.

The drift was defined as:

$$\mathbf{f}(\mathbf{x}, \mathbf{u}, t; \boldsymbol{\xi}) = \begin{bmatrix} \dot{\boldsymbol{\theta}} \\ \mathcal{M}^{-1}(\boldsymbol{\theta})(\boldsymbol{\tau}_m(\boldsymbol{\theta}, \dot{\boldsymbol{\theta}}, \mathbf{u}) - \mathcal{C}(\boldsymbol{\theta}, \dot{\boldsymbol{\theta}})\dot{\boldsymbol{\theta}} + \tau_p(\boldsymbol{\theta}, \dot{\boldsymbol{\theta}}; \xi)) \end{bmatrix}, \tag{32}$$

where $\boldsymbol{\theta} = (\theta_1, \theta_2)$, $\dot{\boldsymbol{\theta}} = (\dot{\theta}_1, \dot{\theta}_2)$, $\mathcal{M}^{-1}$ is the inverse of the inertia matrix, $\mathcal{C}$ is the Coriolis/

centripetal matrix, $\boldsymbol{\tau}_m$ is the joint torque vector produced by muscles and $\boldsymbol{\tau}_p$ is the external disturbance vector.

The joint torque $\boldsymbol{\tau}_m$ was computed from the product of a (constant) moment arm matrix and the muscle force vector, where each muscle consisted of an elastic element and a viscous element arranged in parallel (Kelvin-Voigt model). Each muscle's viscoelastic properties thus depended on its corresponding input $\mathbf{u}_i$, $i \in \{1, \ldots, 6\}$. The reader is referred to the references and the codes for more details about all equations and parameter values.

The external disturbance $\boldsymbol{\tau}_p$ was defined as:

$$\boldsymbol{\tau}_p(\boldsymbol{\theta}, \dot{\boldsymbol{\theta}}; \xi) = \mathbf{J}(\boldsymbol{\theta})^\top \begin{bmatrix} F_\xi \\ 0 \end{bmatrix}, \tag{33}$$

where $\mathbf{J}$ is the Jacobian matrix of the 2-dof arm and $F_\xi = 1.5\xi\dot{y}$ with $\dot{y}$ the forward hand velocity, that is, the second component of the vector $\mathbf{J}(\boldsymbol{\theta})\dot{\boldsymbol{\theta}}$.

Note that the Bernoulli variable here activates a state-dependent force field.

For the internal noise, we considered a 2-dimensional Wiener process associated with the following diffusion term (i.e., an additive motor noise at torque level):

$$\mathbf{G}(\mathbf{x}, \mathbf{u}, t) = \begin{bmatrix} \mathbf{0}_{2\times 2} \\ \mathcal{M}^{-1}(\boldsymbol{\theta})\sigma_a \end{bmatrix}. \tag{34}$$

As a cost function, we considered a mixture of smoothness, energy and variance as follows:

$$J(\mathbf{u}) = 10^4 \mathrm{tr}(\mathbf{J}(\mathbf{m}_\theta(T))\mathbf{P}_\theta(T)\mathbf{J}(\mathbf{m}_\theta(T))^\top) + \int_0^T (\mathbf{u}(t)^\top \mathbf{u}(t) + 0.5\mathbf{a}(t)^\top \mathbf{a}(t)\,dt,$$

where $\mathbf{m}_\theta$ is the 2-D position vector extracted from the mean state $\mathbf{m} = \mathbb{E}[\mathbf{x}]$, $\mathbf{P}_\theta$ is the corresponding 2x2 positional covariance matrix extracted from $\mathbf{P} = \mathbb{E}[(\mathbf{x} - \mathbf{m})(\mathbf{x} - \mathbf{m})^\top]$, and $\mathbf{a} = \dot{\mathbf{J}}(\mathbf{m}_\theta)\mathbf{m}_{\dot{\theta}} + \mathbf{J}(\mathbf{m}_\theta)\mathbf{m}_{\ddot{\theta}}$ is the Cartesian hand acceleration with $\mathbf{m}_{\dot{\theta}}$, $\mathbf{m}_{\ddot{\theta}}$ the 2-D vectors extracted from $\dot{\mathbf{m}} = \mathbf{f}(\mathbf{m}, \mathbf{u}, t; \boldsymbol{\mu})$. Hence the above cost function only depends on the mean and covariance of the random process $\mathbf{x}$ and control $\mathbf{u}$ as in Eq 5.

In our simulations, the initial state covariance was set to $\mathbf{P}_0 = 10^{-6}\mathbf{I}$, the additive noise was $\sigma_a = 0.025$ and the duration $T = 0.75s$.

## Modeling of the uncertain wrist reaching experiment

We modeled the drift term $\mathbf{f}$ by the following system:

$$\begin{cases} I\ddot{\theta} &= \tau_1 - \tau_2 - b\dot{\theta} + \tau_p(t)\xi, \\ \dot{a}_1 &= (u_1 - a_1)/\rho, \\ \dot{a}_2 &= (u_2 - a_2)/\rho, \end{cases} \tag{35}$$

where $u_1, u_2 \in [0, 1]$ are the muscle inputs of the flexor and extensor carpi radialis respectively, $a_1, a_2$ are the corresponding muscle activations, $\tau_1, \tau_2$ are the corresponding muscle torques and $\theta, \dot{\theta}$ are the joint angle/velocity. The parameters $I = 0.01$ kg.m$^2$ and $b = 0.05$ Nm.s/rad are respectively the moment of inertia about the wrist and the joint's viscosity, and $\rho = 0.04$ s is the response time for muscle activations. As before, $\xi$ was a Bernoulli variable with probability $\alpha$ and $\tau_p$ was defined as in Eq 26 with $\tau_p(t) = \tau_e(t - 0.8T)$ where $T$ is the movement duration. This 0.8 value was taken from experimental data which showed that the disturbance was

activated around 80% of the total duration on average. Note that the Bernoulli variable here activates a time-dependent force field.

The muscle torques were defined as in [3] to capture their variable viscoelasticity:

$$\begin{cases} \tau_1 & = & a_1[k_n - k_s(\theta - \theta_r) - k_d(\dot{\theta} - \dot{\theta}_r)], \\ \tau_2 & = & a_2[k_n + k_s(\theta - \theta_r) + k_d(\dot{\theta} - \dot{\theta}_r)]. \end{cases} \tag{36}$$

In the above equation, $\theta_r, \dot{\theta}_r$ were taken from a reference trajectory built from a minimum jerk solution [43] and we set $k_n = 15$ Nm, $k_s = 15$ Nm/rad and $k_d = 1.5$ Nm/rad/s based on order of magnitudes found in the literature [39, 82, 83].

By denoting the state as $\mathbf{x} = (\theta, \dot{\theta}, a_1, a_2)$ and the control as $\mathbf{u} = (u_1, u_2)$, the system of Eq 35 can be written as a drift $\mathbf{f}(\mathbf{x}, \mathbf{u}, t; \xi)$ in the form of Eq 1.

The diffusion term was modeled by the matrix:

$$\mathbf{G}(\mathbf{x}, \mathbf{u}, t) = \begin{bmatrix} 0 & 0 & 0 & 0 \\ 0 & 0 & 0 & 0 \\ \sigma_a & \dfrac{\sigma_m u_1}{\rho} & 0 & 0 \\ 0 & 0 & \sigma_a & \dfrac{\sigma_m u_2}{\rho} \end{bmatrix}, \tag{37}$$

where $\sigma_a = 0.02\%$ and $\sigma_m = 2\%$ represent the magnitudes of additive and multiplicative noise respectively. These values were chosen such that the predicted co-contraction in the 0% condition and the rate of success overall matched the experimental values.

The task was to move the wrist from an initial random state $\mathbf{x}(0) = \mathbf{x}_0 \sim \mathcal{N}(\mathbf{m}_0, \mathbf{P}_0)$ with $\mathbf{m}_0 = (-\pi/6, 0, 10^{-4}, 10^{-4})$ and $\mathbf{P}_0 = 10^{-5}\mathbf{I}$ to $\mathbf{m}_T = (\pi/6, 0, \cdot, \cdot)$ where $\cdot$ stands for undefined/free values.

The objective was to minimize the expectation of a quadratic expected cost mixing error and effort terms as follows:

$$J(\mathbf{u}) = \mathbb{E}\left[(\theta(T) - \pi/6)^2 + 0.1\,\dot{\theta}(T)^2 + \int_0^T \left(u_1(t)^2 + u_2(t)^2\right) dt\right]. \tag{38}$$

## Numerical solution to the USOOC problem

In all cases, the associated deterministic optimal control problems were solved numerically using Julia 1.10 and the packages Symbolics [84], JuMP [85] and Ipopt [86]. More precisely, we generated the augmented dynamics using symbolic calculations. Thanks to JuMP, we then transcribed the continuous problem into a NLP problem using a trapezoidal scheme for the discretized dynamics [87]. Note that the Cholesky decomposition was used to rewrite the covariance differential equation as in [88], so that simple box constraints on the augmented state could be used to ensure the positive definiteness of the covariance matrix along the trajectory. Finally, using the automatic differentiation feature of JuMP, we employed the Ipopt solver for the NLP problem. Initial guesses were generated from simpler instances of the problem (e.g., without uncertainty) and reused for problems with uncertainties. All figures were made with Makie [89]. The code used to generate all simulations and figures, together with the data, is available as a S1 Data.

## Supporting information

**S1 Data. Simulation codes and data sets to create the figures.**
(ZIP)

**S1 Text. Additional details for discrete random variables and mathematical proofs.**
(PDF)

## Author Contributions

**Conceptualization:** Bastien Berret, Frédéric Jean.

**Formal analysis:** Bastien Berret, Dorian Verdel, Frédéric Jean.

**Funding acquisition:** Bastien Berret.

**Investigation:** Dorian Verdel.

**Methodology:** Bastien Berret, Frédéric Jean.

**Project administration:** Bastien Berret.

**Software:** Bastien Berret, Dorian Verdel.

**Supervision:** Bastien Berret, Etienne Burdet.

**Visualization:** Bastien Berret, Dorian Verdel.

**Writing – original draft:** Bastien Berret, Dorian Verdel, Frédéric Jean.

**Writing – review & editing:** Bastien Berret, Dorian Verdel, Etienne Burdet, Frédéric Jean.

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
