## [Decision Letter · Decision Letter 0]

19 Aug 2024

Dear Dr. Berret,

Thank you very much for submitting your manuscript "Co-Contraction Embodies Uncertainty: An Optimal Feedforward Strategy for Robust Motor Control" for consideration at PLOS Computational Biology.

As with all papers reviewed by the journal, your manuscript was reviewed by members of the editorial board and by several independent reviewers. The reviewers were generally positive about the contribution of the paper, but raised a few issues that should be addressed, as well as making a number of constructive suggestions to improve the paper. In light of the reviews (below this email), we would like to invite the resubmission of a revised version that takes into account the reviewers' comments.

We cannot make any decision about publication until we have seen the revised manuscript and your response to the reviewers' comments. Your revised manuscript is also likely to be sent to reviewers for further evaluation.

Sincerely,

Adrian M Haith

Academic Editor

PLOS Computational Biology

Andrea E. Martin

Section Editor

PLOS Computational Biology

Reviewer's Responses to Questions

**Comments to the Authors:**

Reviewer #1: Review of “Co-Contraction Embodies Uncertainty: An Optimal Feedforward Strategy for Robust Motor Control” by Berret et al.

This paper presents theoretical analysis and experimental evidence that feedforward modulation of muscle co-activation and mechanical impedance is a strategy to deal with uncertain external perturbations. The paper is generally well written and the experimental results are largely consistent with the model predictions.

However, the experiment considered only a single-joint pointing task which is hardly a “seminal motor control task” (line 89). This significantly reduced enthusiasm for the findings reported.

Moreover, the finding that for \\alpha of 0.25, 0.50 and 0.75, performance converged within 10 to 20 trials (figure 5C) would appear to indicate that this was not a particularly challenging task. How then can extrapolation to more general tasks be justified?

It appears (equation 1) that the external disturbance process \\xi was stationary and independent of the system state. While that is an important case, it seems like a rather impoverished model of tool use, where the user’s actions significantly affect the stochasticity of the interaction as well as its stability.

Indeed, due to the apparent absence of state-dependent disturbances, it is not clear how equation (1) was applied to the simulation of figure 3. That may reflect this reviewers ignorance of the mathematics but if so, this should be clarified in the text.

The results presented in figure 2 are interesting but they indicate an approximately trapezoidal time profile of stiffness. How can this be reconciled with the results reported by Bennett et al. (ref. 31) which clearly showed a substantial reduction of stiffness in the middle of movement? Indeed, might Bennett et al.’s results invalidate the analysis presented here? This should at least be discussed briefly.

The paper is focused on stiffness modulation but neuromuscular mechanical impedance may have other components. Was there any evidence of modulation of apparent neuromuscular damping?

I would like to know more about the exoskeleton. How “back-drivable” was it? For example, if its static friction or apparent inertia were significant (which is typical for a geared electric motor) they would evoke enhanced muscle activation and thereby enhance apparent mechanical impedance with no relation to external perturbations.

Minor suggestions:

The use of the symbol u to represent either (i) generic control input (ii) torque and (iii) stiffness (e.g. in figure 2) was quite confusing; torque and stiffness require different units. It would be helpful to distinguish these different meanings of this variable.

Abstract line 7, Line 108, Line 268: Do you mean “singular” in the mathematical sense or do you mean “unique”? If the former, please clarify.

Line 13: re-arrange the sentence to read: “… predict co-contraction well …”

Line 21: while direct measurement of output mechanical impedance requires external perturbation, it may also be estimated from a model. Remarkably, there is evidence that humans do so [1], [2] . This might support the argument for predictive feedforward control of mechanical impedance.

In equation (1) should the symbol be \\w_t or w as defined in the subsequent text?

Line 259: delete “of”

Line 236: suggest replacing “At last” with “Finally”.

Line 454: rearrange the sentence to read: “… the considered reaching movement.”

[1] M. E. Huber, C. Folinus, and N. Hogan, “Visual Perception of Joint Stiffness from Multi-Joint Motion,” J. Neurophysiol., vol. 122, no. 1, pp. 51–59, 2019, doi: 10.1152/jn.00514.2018.

[2] A. M. J. West, M. E. Huber, and N. Hogan, “Role of Path Information in Visual Perception of Joint Stiffness,” PLoS Comput. Biol., vol. 18, no. 11, p. e1010729, 2022, doi: 10.1371/journal.pcbi.1010729.

Reviewer #2: This paper presents a new mathematical framework to explain human sensorimotor behavior in the face of environmental uncertainty. The framework and the experimental evaluations focus on feedforward (open-loop) control, showing that muscle co-contraction is related to environmental uncertainty. The results provide an insightful first step towards understanding why and how co-contraction occurs, and can be a valuable addition to the literature. However, I have some general thoughts that the authors might consider addressing to further strengthen the paper.

One point for further consideration is the potentially restrictive formulation of the problem. The authors model “trial-by-trial” uncertainty, where the uncertain effects are fixed within the scope of one trial (encoded in \\xi), but we live in a continuous world; trial-by-trial effects are artifacts of our experimental designs. I recognize the limitations of our current experimental and theoretical tools, and that answering such real-world questions will require decades of research. Nevertheless, I encourage the authors to discuss this issue further. I personally would be very interested to learn their perspective on how these results might be extended to more real-world applications, such as implementing the framework within a receding horizon control scheme.

My second comment is regarding the level of detail. The paper presents the general problem that the nervous system may need to solve (eqs. 1 and 2) and then converts the original problem to an equivalent ODE problem. However, the final step—solving the problem—is not fully detailed. Depending on the reader's goal—whether learning about the functioning of the brain or implementing/extending the method—the level of detail might be either too much or too little. Adjusting the level of detail to better match the overall objective of the paper would be beneficial.

Another philosophical point concerns the paper's conclusion: “Co-contraction embodies uncertainty.” The presented optimal feedforward control model indeed supports this conclusion. However, a short discussion on whether this is the sole reason for co-contraction in uncertain environments might be beneficial. The authors’ 2020 SOOC paper is a good example; it provided an alternative explanation for some behavioral observations that were previously modeled differently, e.g., the minimal intervention principle. Could there be alternative explanations for co-contraction yet to be discovered (maybe a feedback-driven one)?

Lastly, the manuscript is well-written, though a round of copy-editing could enhance its clarity and impact.

Here are some additional comments

Line 63. Problem 1. The dynamics of the nonlinear model must be approximated (linearized) to arrive at this augmented-state ODE description. However, the Discussion (lines 253-254) tries to contrast the model with “linear” theories like H-infinity. Further discussion is appreciated to expand on when and how the presented method can be applied to full nonlinear system equations (i.e., without approximation), or how the approximation is still superior to other methods.

Equations 6, 7, 10, and 25 need parentheses in the integral.

115. How was the multiplicative noise included in the model? Please provide more details.

Fig 1C. Why do the grey traces have two separate values for a given variance? In other words, why isn’t the shape shown in Fig 1A symmetric? Is it because of some combined effects of intrinsic and extrinsic noise?

120. A brief description of the model is appreciated here. Is it a linear or nonlinear model? How were stiffness and muscle activity related? I suggest including the equations of motion in the Methods section.

136. Fig 3 or Fig 4?

141. Please provide more details of this model as well.

150. Torques are mentioned, but there are 3 traces in Fig 3C,D, which implies muscle groups. Please provide further details to clarify.

151. In the statement “mean stiffness and co-contraction”: Mean over time and over muscles? Further details are appreciated.

Figure 3C. It is curious that co-contraction is zero in the first quarter of movement. Is it just an artifact of the combination of reach direction, initial condition, disturbance magnitude, etc. in the simulation? Or is there some exciting explanation behind it?

165. Was the disturbance always in the flexion direction (or extension)? Also, when describing the experimental protocol, it is mentioned in line 410 both flexion and extension movements, but the results (and line 421) imply only flexion movements were tested. The description of the results and experimental protocol require further expansion and clarification.

303. Why are A, B assumed to be functions of u? Is it, for example to include muscle activation-dependent dynamics? What would be the applications/implications of this assumption?

**Have the authors made all data and (if applicable) computational code underlying the findings in their manuscript fully available?**

Reviewer #1: Yes

Reviewer #2: Yes

PLOS authors have the option to publish the peer review history of their article (what does this mean?). If published, this will include your full peer review and any attached files.

Reviewer #1: No

Reviewer #2: **Yes: **Reza Sharif Razavian
---

## [Decision Letter · Decision Letter 1]

1 Oct 2024

Dear Dr. Berret,

Thank you very much for submitting your manuscript "Co-Contraction Embodies Uncertainty: An Optimal Feedforward Strategy for Robust Motor Control" for consideration at PLOS Computational Biology.

The reviewers were satisfied with the responses and revisions to the paper. However, Reviewer 1 noted a number of small issues that require attention before formal acceptance.

Sincerely,

Adrian M Haith

Academic Editor

PLOS Computational Biology

Andrea E. Martin

Section Editor

PLOS Computational Biology

Reviewer's Responses to Questions

**Comments to the Authors:**

Reviewer #1: Please see the attached review

Reviewer #2: Thank you for addressing my comments. I have no further comments.

**Have the authors made all data and (if applicable) computational code underlying the findings in their manuscript fully available?**

Reviewer #1: Yes

Reviewer #2: Yes

PLOS authors have the option to publish the peer review history of their article (what does this mean?). If published, this will include your full peer review and any attached files.

Reviewer #1: No

Reviewer #2: **Yes: **Reza Sharif Razavian

Figure Files:

Data Requirements:

Reproducibility:

References:

---

## [Editor Report · Decision Letter 2]

29 Oct 2024

Dear Dr. Berret,

We are pleased to inform you that your manuscript 'Co-Contraction Embodies Uncertainty: An Optimal Feedforward Strategy for Robust Motor Control' has been provisionally accepted for publication in PLOS Computational Biology.

Best regards,

Adrian M Haith

Academic Editor

PLOS Computational Biology

Andrea E. Martin

Section Editor

PLOS Computational Biology

Feilim Mac Gabhann

Editor-in-Chief

PLOS Computational Biology

Jason Papin

Editor-in-Chief

PLOS Computational Biology

---

## [Editor Report · Acceptance letter]

14 Nov 2024

PCOMPBIOL-D-24-01022R2 

Co-Contraction Embodies Uncertainty: An Optimal Feedforward Strategy for Robust Motor Control

Dear Dr Berret,

I am pleased to inform you that your manuscript has been formally accepted for publication in PLOS Computational Biology. Your manuscript is now with our production department and you will be notified of the publication date in due course.

With kind regards,

Anita Estes
